# RNA-binding protein ZFP36L1 regulates osteoarthritis by modulating members of the heat shock protein 70 family

Young-Ok Son [1], Hyo-Eun Kim[1], Wan-Su Choi[1], Churl-Hong Chun[2] & Jang-Soo Chun[1]

Osteoarthritis (OA) is a whole-joint disease characterized by cartilage destruction and other whole-joint pathological changes. There is currently no effective disease-modifying therapy. Here we investigate the post-transcriptional mRNA regulation of OA-modulating proteins in chondrocytes and show that the ZFP36 family member, ZFP36L1, is specifically upregulated in OA chondrocytes and OA cartilage of humans and mice. Adenovirus-mediated over-expression of ZFP36L1 alone in mouse knee-joint tissue does not modulate OA pathogenesis. However, genetic ablation or silencing of Zfp36l1 significantly abrogates experimental OA in mice. Knockdown of Zfp36l1 increases the mRNA expression of two heat shock protein 70 (HSP70) family members, which act as its direct targets. Furthermore, overexpression of HSPA1A in joint tissues protects mice against experimental OA by inhibiting chondrocyte apoptosis. Our results indicate that the RNA-binding protein, ZFP36L1, regulates HSP70 family members that appear to protect against OA pathogenesis by inhibiting chondrocyte apoptosis.

[1] National Creative Research Initiatives Center for Osteoarthritis Pathogenesis and School of Life Sciences, Gwangju Institute of Science and Technology, Gwangju 61005, Republic of Korea. [2] Department of Orthopedic Surgery, Wonkwang University School of Medicine, Iksan 54538, Republic of Korea. These authors contributed equally: Young-Ok Son, Hyo-Eun Kim.  Correspondence and requests for materials should be addressed to J.-S.C. (email: jschun@gist.ac.kr)

Osteoarthritis (OA) carries a large socioeconomic cost and stands as a leading cause of disability[1], but we currently lack an effective disease-modifying therapy. OA is associated with multiple pathological changes in whole-joint tissues, including cartilage destruction, synovial inflammation, osteophyte formation, subchondral bone sclerosis, etc.[2,3]. The cartilage destruction that is a hallmark of OA pathogenesis is caused by the upregulation of matrix-degrading enzymes and/or the downregulation of cartilage extracellular matrix (ECM) molecules[4]. Among the matrix-degrading enzymes, matrix metalloproteinase 3 (MMP3), MMP13, and a disintegrin-like and metallopeptidase with thrombospondin type 1 motif 5 (ADAMTS5; aggrecanase-2) are known to play crucial roles in OA cartilage destruction[5–7]. Modulation of SRY (sex-determining region Y)-Box 9 (SOX9), a transcription factor known to regulate type II collagen and aggrecan, is a critical step in the OA-related downregulation of ECM molecules[8]. The apoptosis of chondrocytes also contributes to downregulating cartilage ECM molecules[9]. Matrix-degrading enzymes and ECM molecules are regulated by various extracellular catabolic regulators, including the pro-inflammatory cytokines, interleukin (IL)-1β, IL-6, and tumor necrosis factor (TNF)-α[10]. We previously identified cellular catabolic mediators in chondrocytes, such as the transcription factor, hypoxia-inducible factor (HIF)-2α[11], and the zinc importer, ZIP8 (ref. [12]). Importantly, we showed that these cellular mediators play crucial catabolic functions in OA pathogenesis by upregulating matrix-degrading enzymes and/or downregulating ECM molecules in joint articular chondrocytes[11,12].

The expression of OA-related catabolic and anabolic factors in chondrocytes can be regulated at multiple steps, including via the post-transcriptional regulation of mRNA stability, which is a crucial aspect of gene expression. Regulation of mRNA stability is a complex process that involves the controlled interactions of microRNAs or RNA-binding proteins with target mRNAs. RNA-binding proteins bind directly to AU-rich elements (AREs) in the 3′-untranslated regions (3′-UTRs) of their target mRNAs and promote poly-A tail removal and subsequent mRNA decay[13]. To explore whether the expression levels of OA-related catabolic and anabolic factors in chondrocytes are regulated by RNA-binding proteins, we initially used a microarray analysis to screen for RNA-binding proteins whose expression levels are modulated in chondrocytes subjected to stimulation of OA-associated catabolic signaling, such as by IL-1β treatment[10] or overexpression of ZIP8 or HIF-2α[11,12]. In our initial screening analysis, we found that a member of the ZFP36 (zinc-finger protein 36) family of RNA-binding proteins was specifically upregulated in chondrocytes subjected to stimulation of OA-associated catabolic signaling. ZFP36 is an ancient RNA-binding protein that is also known as tristetraprolin (TTP) and TPA-inducible sequence 11 (TIS11). The ZFP36 family of RNA-binding proteins consists of ZFP36 (TTP, TIS11), ZFP36L1 (BRF-1, TIS11b), ZFP36L2 (BRF-2, TIS11d), and ZFP36L3 (only in rodents)[14–17]. These proteins contain two tandemly repeated zinc-finger motifs (CCCH), which bind to AREs in the 3′-UTRs of target mRNAs to trigger ARE-mediated mRNA destabilization. The ZFP36 family members are structurally similar, but they all show cell-type-specific expression patterns and play distinct cellular functions, probably by regulating different target mRNAs[14–17]. For instance, ZFP36 acts as a signal-regulated switch during immune-cell activation[18], ZFP36L1 maintains bile acid levels in hepatocytes[19], and ZFP36L2 regulates the E2F pathway during cell-cycle progression[20]. The ZFP36 family members may also have isoform-specific functions, as knockout (KO) mice show isoform-specific and non-overlapping phenotypes[14]. For example, Zfp36 KO mice were observed to be normal at birth, but soon developed a severe inflammatory syndrome with cachexia and autoimmunity owing

to excessive activity of TNF-α[21]. In contrast, disruption of Zfp36l1 resulted in lethality by embryonic day 11, apparently because the allantois failed to fuse with the chorion and umbilical circulation never initiated[22]. Finally, Zfp36l2-KO mice appeared normal but exhibited complete female infertility due to dysregulation of the luteinizing hormone receptor, which resulted in embryonic failure at approximately the two-cell stage[23].

The various members of the ZFP36 family are known to regulate both common and distinct mRNA targets[16,17]. However, no previous study has examined whether they regulate the mRNAs involved in OA pathogenesis. Here, we investigated the possible association of ZFP36 family members with OA pathogenesis. We found that ZFP36L1 is specifically upregulated in OA chondrocytes and OA cartilage of humans and mice. We then evaluated the functions of ZFP36L1 in OA pathogenesis and examined the relevant regulatory mechanisms. We report here that although adenovirus-mediated overexpression of ZFP36L1 in mouse knee-joint tissue did not modulate OA pathogenesis, genetic ablation or silencing of Zfp36l1 significantly abrogated experimental OA in mice subjected to destabilization of the medial meniscus (DMM). We also show that ZFP36L1 regulates the mRNAs encoding two members of the heat-shock protein 70 (HSP70) family that appear to exert protective effects on OA pathogenesis by inhibiting chondrocyte apoptosis.

## Results

**ZFP36L1 is upregulated in OA chondrocytes and cartilage.** To explore the possible association of RNA-binding proteins with OA pathogenesis, we first analyzed the expression levels of RNA-binding proteins in primary-culture chondrocytes subjected to stimulation of catabolic signaling by treatment with the pro-inflammatory cytokine, IL-1β[10], or adenovirus-mediated overexpression of HIF-2α (Ad-HIF-2α)[11] or the zinc importer, ZIP8 (Ad-ZIP8)[12]. We used passage 0 mouse chondrocytes, having first confirmed that they exhibited the expected differentiated phenotypes, such as high-level expression of type II collagen and low-level expression of type I collagen (Supplementary Fig. 1a). Our microarray analysis revealed that, from among the examined RNA-binding proteins, a member of the ZFP36 family was specifically increased in chondrocytes stimulated with catabolic regulators (Fig. 1a; Supplementary Table 1). To validate and expand upon this finding, we characterized the expression levels of ZFP36 family members in OA cartilage and primary-culture chondrocytes stimulated with the catabolic regulators. Our reverse transcription-polymerase chain reaction (RT-PCR) analysis revealed that ZFP36L1 was specifically upregulated by IL-1β treatment or infection of Ad-HIF-2α or Ad-ZIP8 in primary-culture chondrocytes (Supplementary Fig. 1b, c). Quantitative RT-PCR (qRT-PCR) analysis further confirmed that the mRNA level of ZFP36L1 was significantly increased in these chondrocytes (Fig. 1b, c). We then analyzed the levels of ZFP36L1 in OA cartilage from human patients and various mouse models. The protein level of ZFP36L1 was markedly elevated in OA-affected, damaged regions of human cartilage, compared with undamaged areas from the same patient (Fig. 1d). ZFP36L1 protein levels were also markedly increased in chondrocytes of mouse OA cartilage caused by DMM surgery (Fig. 1e, left) or by overexpression of HIF-2α or ZIP8 via intra-articular (IA) injection of Ad-HIF-2α or Ad-ZIP8, respectively[11,12] (Fig. 1e, right).

**ZFP36L1 overexpression does not modulate OA pathogenesis.** Because the above results suggested that ZFP36L1 might play a role in OA pathogenesis, we examined the in vivo functions of ZFP36L1 in OA pathogenesis by overexpressing ZFP36L1 in knee-joint tissues via IA injection of Ad-ZFP36L1 in 12-week-old

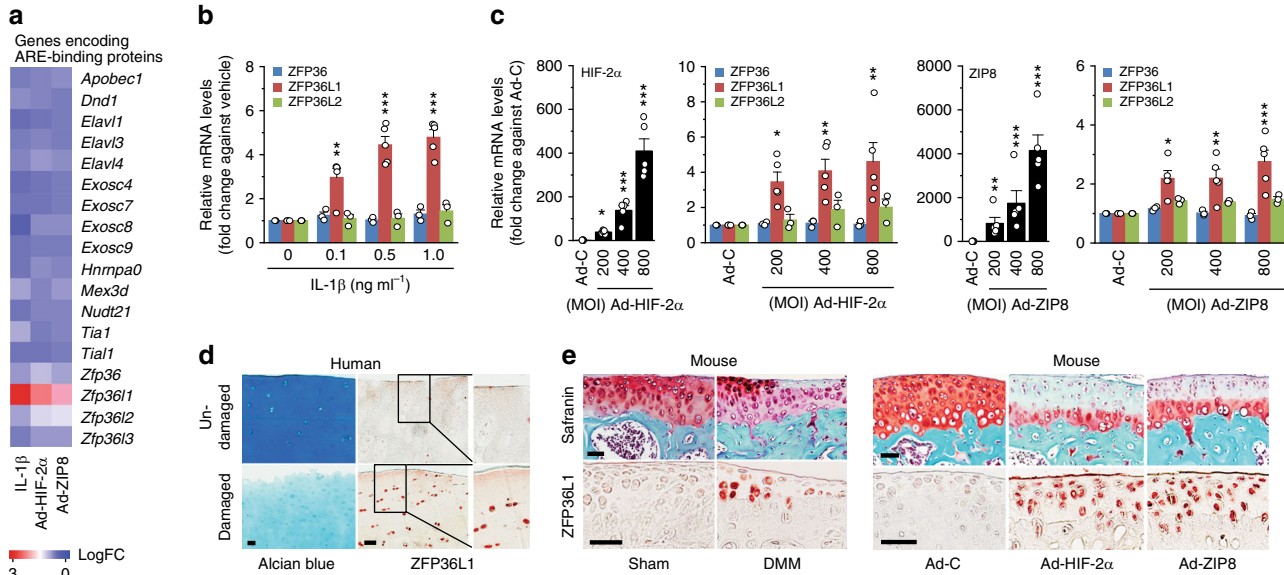

**Fig. 1** Upregulation of ZFP36L1 in chondrocytes stimulated with OA-associated catabolic regulators. **a** Microarray analysis of AU-rich element (ARE)-binding proteins in chondrocytes treated with IL-1β or infected with Ad-HIF-2α or Ad-ZIP8. **b, c** qRT-PCR analyses of ZFP36 family members in chondrocytes treated with IL-1β or infected with 800 MOI of control virus (Ad-C) or the indicated MOIs of Ad-HIF-2α or Ad-ZIP8 ($n = 3$ or 5). **d** Representative images of Alcian blue staining and ZFP36L1 immunostaining in undamaged and damaged regions of OA cartilage from the same patient ($n = 7$ patients). **e** Representative images of Safranin-O staining and ZFP36L1 immunostaining in mouse OA cartilage caused by DMM surgery ($n = 5$ mice per group) or by IA injection of Ad-HIF-2α or Ad-ZIP8 ($n = 5$ mice per group). Means ± s.e.m. with one-way ANOVA (*$P < 0.05$, **$P < 0.005$, and ***$P < 0.0005$). Scale bar: 50 μm

male mice. Our previous work had extensively demonstrated that an adenovirus system can effectively deliver target genes to cartilage and other joint tissues[11,12,24–26]. Consistent with these findings, immunohistochemical staining indicated that three weekly IA injections of Ad-ZFP36L1 triggered the effective overexpression of ZFP36L1 in cartilage, meniscus, synovium, and ligament (Fig. 2a; Supplementary Fig. 2a). However, ZFP36L1 overexpression did not cause cartilage destruction or synovitis, as examined by Safranin-O and H&E staining, respectively (Fig. 2b), or alter the expression levels of the matrix-degrading enzymes, MMP3 and MMP13, in cartilage tissue (Fig. 2a). Similarly, ZFP36L1 overexpression in primary-culture chondrocytes did not affect the expression levels of MMP3, MMP13, and ADAMTS5 which play crucial roles in OA cartilage destruction[5–7], nor did it affect cartilage ECM molecules, such as type II collagen and aggrecan, or the regulatory transcription factor, SOX9 (ref. [8]) (Fig. 2c; Supplementary Fig. 2b). These results collectively indicate that overexpression of ZFP36L1 alone is not sufficient to cause OA pathogenesis in mice. We further examined the effects of ZFP36L1 overexpression in joint tissues of mice subjected to DMM-induced OA pathogenesis. Although ZFP36L1 overexpression via IA injection of Ad-ZFP36L1 tended to enhance the manifestations of OA, as assessed by OARSI grade, osteophyte maturity, and subchondral bone plate thickness, the effects were not statistically significant (Fig. 2d, e). The cartilage destruction caused by the overexpression of HIF-2α or ZIP8 via IA injection of Ad-HIF-2α[11] or Ad-ZIP8 (ref. [12]), respectively, was not affected by the IA injection of Ad-ZFP36L1 (Supplementary Fig. 3a, c). Furthermore, overexpression of ZFP36L1 in primary-culture chondrocytes did not alter the effects of HIF-2α or ZIP8 overexpression on the expression levels of catabolic or anabolic molecules (Supplementary Fig. 3b, d).

**Genetic ablation of *Zfp36l1* inhibits OA pathogenesis.** Next, we examined the functions of ZFP36L1 in OA pathogenesis via loss-of-function approaches by generating *Zfp36l1*-knockout (KO)

mice or by knocking down ZFP36L1 in whole-joint tissues via IA injection of an adenovirus expressing shRNA against ZFP36L1 (Ad-shZFP36L1). For KO, mouse *Zfp36l1* was disrupted by an 8-bp insertion and a 42-bp deletion, both in exon 2 (Supplementary Fig. 4a). Because homozygous KO mice (*Zfp36l1*$^{-/-}$) are not viable[22], we used heterozygous *Zfp36l1*$^{+/-}$ mice for our experimental OA studies. *Zfp36l1*$^{+/-}$ mice exhibited marked decreases in the mRNA and protein levels of ZFP36L1 in the chondrocytes of their cartilage tissues (Supplementary Fig. 4b, c). *Zfp36l1*$^{+/-}$ mice exhibited normal skeletal development, as determined by whole-skeleton staining of E18.5 embryos, von Kossa staining of E18.5 metatarsal bones, and Alcian blue/nuclear fast red staining of metatarsal bones from 2-week-old mice (Supplementary Fig. 4d–f). Following these validations, we compared DMM-induced OA manifestations of these mice versus wild-type (WT) littermates. *Zfp36l1*$^{+/-}$ mice exhibited significant reductions in all manifestations of DMM-induced OA, including cartilage erosion, osteophyte formation, and increased subchondral bone plate thickness, which reflects sclerosis of subchondral bone[27] (Fig. 3a, b). These results clearly indicate that *Zfp36l1*$^{+/-}$ mice exhibit protection from DMM-induced OA. To further examine the function of ZFP36L1 in OA pathogenesis, we performed ZFP36L1 knockdown in whole-joint tissues of mice via IA injection of Ad-shZFP36L1. Similar to our findings in *Zfp36l1*$^{+/-}$ mice, knockdown of ZFP36L1 in joint tissues of DMM-operated mice significantly suppressed all examined OA manifestations, including cartilage erosion, osteophyte formation, and subchondral bone plate thickening (Fig. 3c, d). Consistent with this inhibition of OA cartilage erosion, the DMM-induced upregulation of MMP3 and MMP13 proteins in damaged cartilage was markedly abrogated in *Zfp36l1*$^{+/-}$ mice and Ad-shZFP36L1-injected mice (Supplementary Fig. 5a, b).

**HSPA1A and HSPA1B are targets of ZFP36L1.** Next, we explored possible mechanisms through which ZFP36L1 knockdown might inhibit OA pathogenesis. We speculated that this

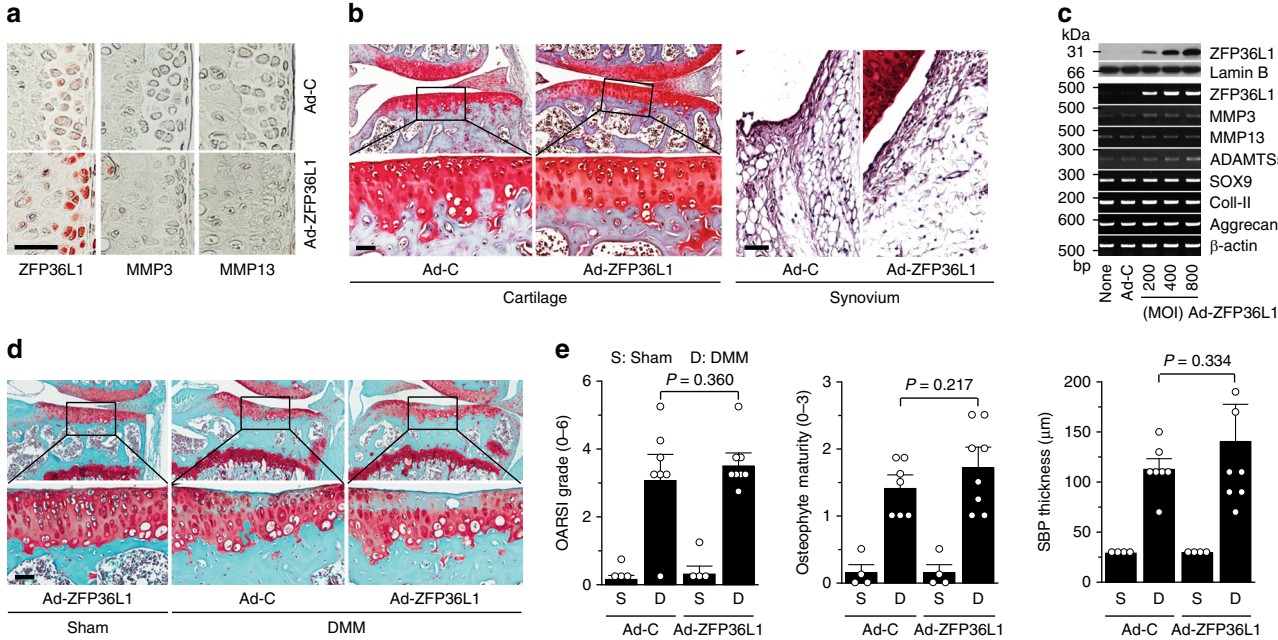

**Fig. 2** Overexpression of ZFP36L1 in joint tissues does not modulate OA pathogenesis. **a**, **b** WT mice were IA injected with Ad-C (n = 6 mice) as a control or Ad-ZFP36L1 (n = 8 mice). Representative immunostaining images of ZFP36L1, MMP3, and MMP13 in cartilage sections (**a**). Cartilage destruction and synovitis were determined by Safranin-O staining and H&E staining 3 weeks after IA injection (**b**). **c** Protein and mRNA levels of the indicated molecules in chondrocytes infected with Ad-C (800 MOI) or the indicated MOIs of Ad-ZFP36L1 for 36 h. Representative images are presented from six independent primary cultures of chondrocytes. **d**, **e** Sham-operated (n = 4 mice) or DMM-operated (n = 7 mice) WT mice were IA injected with Ad-C as a control or Ad-ZFP36L1 to overexpress ZFP36L1 in joint tissues. Representative Safranin-O staining images of joint sections (**d**) and scoring of OARSI grade, osteophyte maturity, and subchondral bone plate thickness (**e**) at 8 weeks after operations. Means ± 95% CI with Mann–Whitney U test for OARSI grade and osteophyte maturity. Means ± s.e.m. with two-tailed t-test for SBP thickness. Scale bar: 50 μm

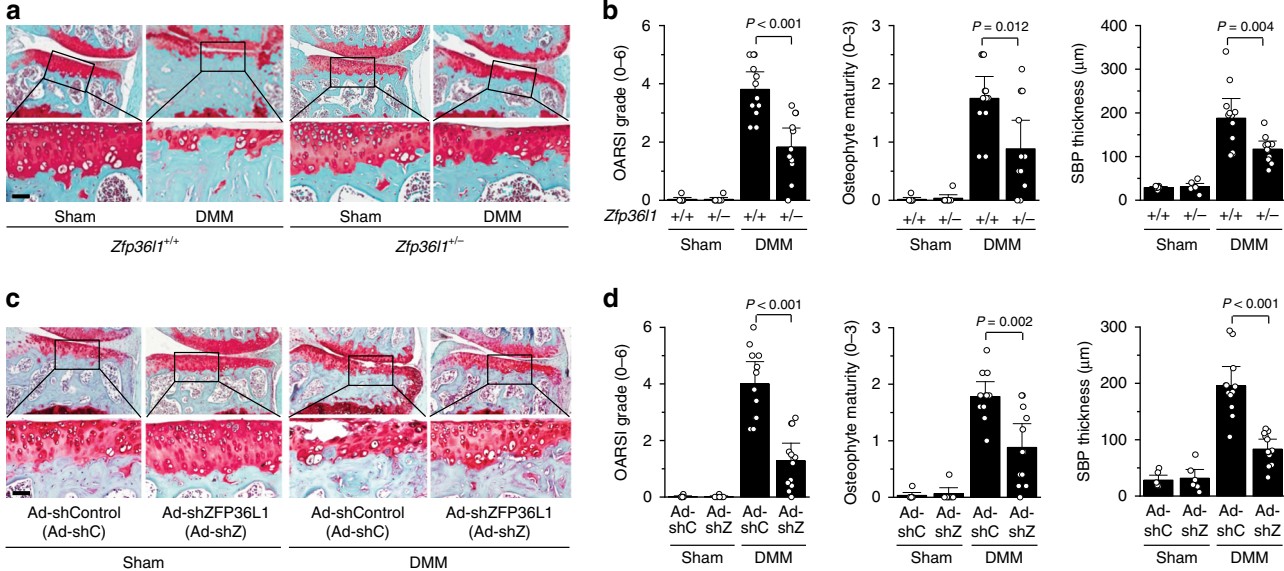

**Fig. 3** Genetic ablation or silencing of *Zfp36l1* abrogates OA pathogenesis in mice. **a**, **b** WT (*Zfp36l1*[+/+]) and *Zfp36l1*[+/−] mice were subjected to sham operation (n = 7 mice per group) or DMM surgery (n = 12 mice per group). Representative Safranin-O staining images (**a**) and scoring of OARSI grade, osteophyte maturity, and subchondral bone plate thickness (**b**). **c**, **d** Sham-operated (n = 6 mice per group) or DMM-operated (n = 12 mice per group) WT mice were IA injected with Ad-shC as a control or Ad-shZFP36L1 to knock down ZFP36L1 in the joint tissues. Representative Safranin-O staining images of joint sections (**c**) and scoring of OARSI grade, osteophyte maturity, and subchondral bone plate thickness (**d**). Means ± 95% CI with Mann–Whitney U test for OARSI grade and osteophyte maturity. Means ± s.e.m. with two-tailed t-test for SBP thickness. SBP subchondral bone plate. Scale bar: 50 μm

could involve increased expression of one or more of the mRNAs targeted by ZFP36L1. We used microarray analysis to examine gene expression profiles in primary-culture chondrocytes subjected to knockdown or overexpression of ZFP36L1 via infection of Ad-shZFP36L1 and Ad-ZFP36L1, respectively (Supplementary Tables 2 and 3). Knockdown of ZFP36L1 increased the levels of various mRNAs; among them, the most marked increases were observed for two members of the HSP70 family, HSPA1A and HSPA1B (Fig. 4a; Supplementary Table 2), which are molecular chaperones that are expressed in response to stress and have been associated with a variety of pathophysiologies[28,29]. Our RT-PCR and western blot analyses further confirmed that the mRNA levels of HSPA1A and HSPA1B and the protein levels of HSP70 were significantly increased by ZFP36L1 knockdown (Fig. 4b, c) and decreased by ZFP36L1 overexpression (Fig. 4c) in primary-culture chondrocytes. However, the mRNA levels of HSPA1A and HSPA1B were not altered by IL-1β treatment (Supplementary Fig. 6a), and these levels were only slightly increased by the overexpression of HIF-2α or ZIP8 (Supplementary Fig. 6a). As HSPA1A and HSPA1B differ by only 10 nucleotides and one amino acid[29], we herein focused on HSPA1A. Our nucleotide sequence analysis identified several ARE motifs in the 3′-UTR of HSPA1A (and HSPA1B) (Supplementary Fig. 6b, c), so we examined whether ZFP36L1 regulates the HSPA1A mRNA as a direct target. We performed RNA-binding protein immunoprecipitation (RIP) assays in chondrocytes. Of the three primer sets arbitrarily designed from the 3′-UTR sequences of ZFP36L1, the first was the most effective in our RIP assays (Fig. 4d; Supplementary Fig. 6b; Supplementary Table 4). Our results revealed that ZFP36L1 bound to the 3′-UTR of the HSPA1A mRNA (Fig. 4d). Consistent with this, overexpression of ZFP36L1 in chondrocytes enhanced HSPA1A mRNA decay and reduced the protein level of HSP70 (Fig. 4e). Our results collectively suggest that ZFP36L1 binds to one or more AREs in the 3′-UTR of HSPA1A to induce ARE-mediated mRNA decay in mouse articular chondrocytes.

**HSPA1A protects against DMM-induced OA pathogenesis.** As members of the HSP70 family exhibit protective effects against inflammation, oxidative stress, and apoptosis in a variety of cell types[28], we examined whether HSPA1A mediates the inhibitory effects of ZFP36L1 knockdown in OA pathogenesis. Toward this end, we overexpressed HSPA1A in knee-joint tissues of 12-week-old male mice via IA injection of an adenovirus expressing HSPA1A (Ad-HSPA1A). Immunohistochemical staining indicated that HSPA1A was effectively overexpressed in various knee-joint tissues of these mice, including cartilage, meniscus, synovium, and ligament (Fig. 5a; Supplementary Fig. 7a). Overexpression of HSPA1A via three weekly IA injections of Ad-HSPA1A did not affect cartilage structure or synovial inflammation in non-OA mice (Fig. 5b). However, HSPA1A overexpression in joint tissues clearly abrogated the cartilage erosion caused by DMM surgery (Fig. 5c, d). DMM surgery also reduced the basal HSP70 protein levels in cartilage tissue (Supplementary Fig. 7b–d). HSP70 protein levels were markedly increased in the cartilage of sham-operated Zfp36l1[+/−] mice or WT mice IA injected with Ad-shZFP36L1 or Ad-HSPA1A; moreover, the elevated levels of HSP70 in cartilage were maintained even after DMM surgery (Supplementary Fig. 7b–d). These results are consistent with the observed chondroprotective effects of HSPA1A. In contrast to the effects on cartilage erosion, HSPA1A overexpression had no significant effect on DMM-induced osteophyte formation or subchondral bone plate thickening (Fig. 5c, d). Our results clearly indicate that the ZFP36L1-regulated HSP70 family member, HSPA1A, exerts inhibitory

effects on cartilage degeneration in post-traumatic experimental OA in mice.

**MMPs and cartilage ECMs are not direct targets of HSPA1A.** To elucidate the mechanisms through which HSPA1A protects against cartilage degeneration during OA pathogenesis, we first examined whether matrix-degrading enzymes and cartilage ECM molecules are direct targets of the ZFP36L1-HSPA1A pathway. We found that MMP3, MMP13, and ADAMTS5, which are essential matrix-degrading factors in cartilage destruction[5–7], each contain at least one ARE sequence in their 3′-UTRs (Supplementary Fig. 8a). Knockdown of ZFP36L1 via infection of Ad-shZFP36L1 abrogated the IL-1β-induced upregulation of these matrix-degrading enzymes (Fig. 6a; Supplementary Fig. 8b), suggesting that they could be direct targets of ZFP36L1. However, overexpression of ZFP36L1 did not affect the IL-1β-induced upregulation of these enzymes (Supplementary Fig. 8b, c). Furthermore, overexpression of HSPA1A did not modulate the expression levels of the matrix-degrading enzymes in the absence or presence of IL-1β (Supplementary Fig. 8d). Similarly, we observed that type II collagen, aggrecan, and their upstream transcription factor, SOX9, also contain ARE sequences in their 3′-UTRs (Supplementary Fig. 8a), but knockdown of ZFP36L1 or overexpression of HSPA1A did not modulate the expression levels of these ECM molecules (Supplementary Fig. 8b–e). Our results indicate that the examined matrix-degrading enzymes and cartilage ECM molecules are not direct targets of the ZFP36L1-HSP70 pathway, further suggesting that the protective effects of ZFP36L1 knockdown or HSPA1A overexpression on OA are not associated with the direct regulation of these catabolic and anabolic effector molecules.

**HSPA1A modulates chondrocyte apoptosis in mice.** The apoptosis of chondrocytes plays a crucial role in OA cartilage destruction[9], and HSPA1A is known to protect various cell types from apoptosis[28]. We thus examined whether HSPA1A protects against chondrocyte apoptosis. Apoptosis of primary-culture chondrocytes was induced by treatment with the nitric oxide (NO) donor, sodium nitroprusside (SNP), as previously described[30]. SNP treatment did, indeed, cause chondrocyte apoptosis, and this was significantly inhibited by the overexpression of HSPA1A via infection of Ad-HSPA1A (Fig. 6b). To examine whether the chaperone function of HSP70 is required for the inhibition of chondrocyte apoptosis, we employed K71E-HSPA1A, which lacks ATPase activity and the chaperone function[31]. Similar to Ad-HSPA1A infection, WT-HSPA1A protected chondrocytes against SNP-induced apoptosis; in contrast, K71E-HSPA1A exerted no effect on chondrocyte apoptosis (Fig. 6c). This is consistent with a previous report that the chaperone function of HSP70 is required to protect a PEER cell line against heat-induced apoptosis[32]. We next tested whether HSPA1A modulates DMM-induced chondrocyte apoptosis in mouse cartilage tissue. Consistent with a previous report[25], most of the detected apoptotic chondrocytes were observed in the calcified cartilage, not the articular cartilage regions, of both sham- and DMM-operated mice (Fig. 6d–f). However, DMM surgery significantly increased the number of apoptotic chondrocytes in articular cartilage regions (Fig. 6d–f). Strikingly, overexpression of HSPA1A in joint tissues via IA injection of Ad-HSPA1A significantly attenuated DMM-induced chondrocyte apoptosis in articular cartilage (Fig. 6d). Similarly, DMM-induced chondrocyte apoptosis in articular cartilage was significantly inhibited in Zfp36l1[+/−] mice (Fig. 6e) or in WT mice IA injected with Ad-shZFP36L1 (Fig. 6f). All of these results are consistent with the ability of HSPA1A overexpression to inhibit DMM-induced

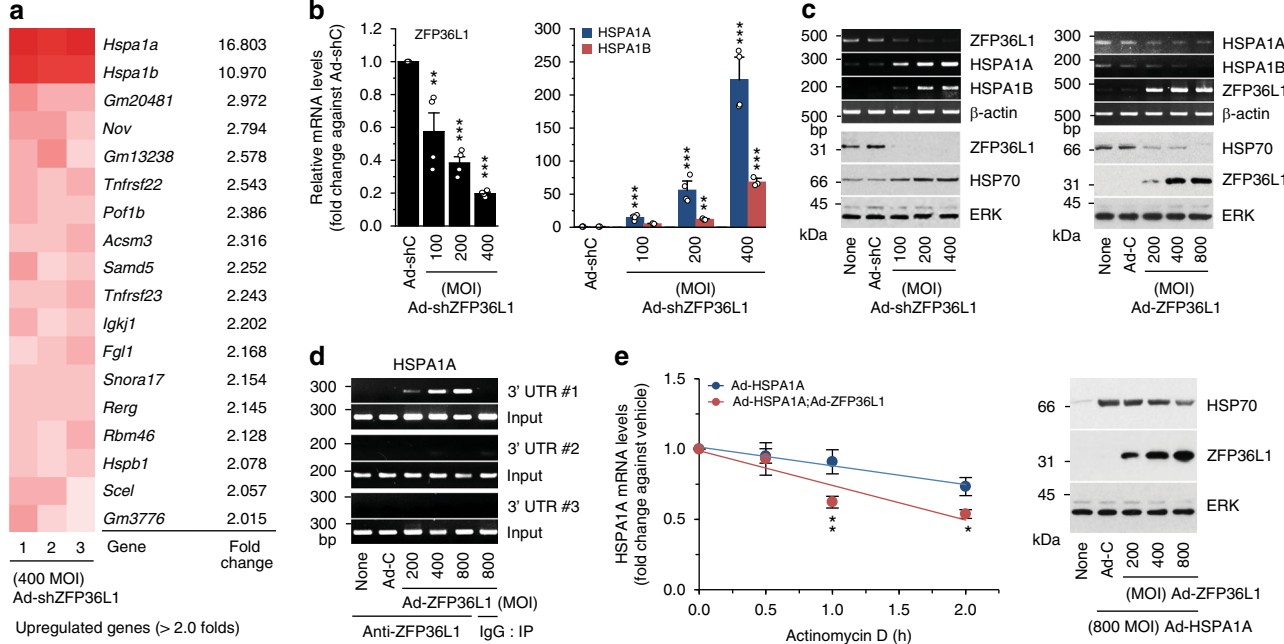

**Fig. 4** ZFP36L1 targets HSP70 family members in chondrocytes. **a** Microarray analysis of chondrocytes infected with 400 MOI of Ad-shZFP36L1 to knock down ZFP36L1 (3 replicates). **b**, **c** qRT-PCR (**b**) and RT-PCR and western blot analyses (**c**) of ZFP36L1, HSPA1A, HSPA1B, and HSP70 in chondrocytes infected with Ad-C (800 MOI) or the indicated MOIs of Ad-shZFP36L1 or Ad-ZFP36L1 for 36 h ($n = 4$). Representative images are presented (**c**). **d** RNA binding of ZFP36L1 in chondrocytes infected with Ad-C or Ad-ZFP36L1. The binding of ZFP36L1 to the 3′-UTR of HSPA1A was determined by RT-PCR of immunoprecipitates obtained using anti-ZFP36L1 or IgG. Representative images are presented from five biologically independent samples. **e** mRNA decay assay. Chondrocytes were infected with Ad-HSPA1A with or without Ad-ZFP36L1 for 12 h, and then exposed to Actinomycin D (1 μg ml$^{-1}$) for the indicated time periods. The mRNA levels of HSPA1A were quantified by qRT-PCR analysis ($n = 9$). Representative western blot images of HSP70 and ZFP36L1 proteins. Means ± s.e.m. with one-way ANOVA (*$P < 0.01$, **$P < 0.001$, and ***$P < 0.0001$).

cartilage destruction. Our results collectively indicate that a ZFP36L1-modulated HSP70 family member inhibits chondrocyte apoptosis both in vitro and in vivo, and thereby contributes to inhibiting OA cartilage destruction.

**Knockdown of HSPA1A abrogates the ability of ZFP36L1 in OA.** Finally, we examined whether inhibiting the upregulation of HSPA1A in cartilage tissue is sufficient to restore the OA phenotype in $Zfp36l1^{+/-}$ mice subjected to DMM surgery. Toward this end, we knocked down HSPA1A in $Zfp36l1^{+/-}$ mice and WT littermates via IA injection of Ad-shHSPA1A in sham- or DMM-operated mice. We found that IA injection of Ad-shHSPA1A effectively inhibited the increase of HSP70 protein levels in the cartilage of DMM-operated mice (Fig. 7a). Consistent with this, knockdown of HSPA1A in $Zfp36l1^{+/-}$ mice abolished the inhibitory effects of $Zfp36l1$ KO on cartilage destruction in DMM-operated mice (Fig. 7a, b). In contrast to the effects of HSPA1A knockdown on cartilage erosion, this treatment had no significant effect on DMM-induced osteophyte formation or subchondral bone plate thickening (Fig. 7a, b). These results clearly support our contention that upregulation of HSPA1A is responsible for the inhibitory effects of ZFP36L1 knockout on DMM-induced cartilage destruction. Additionally, our results indicate that HSPA1A mediates cartilage erosion, but not osteophyte formation or subchondral bone plate thickening. This is consistent with the above observations that HSPA1A overexpression abrogated DMM-induced cartilage erosion but did not alter osteophyte formation or subchondral bone plate thickening in these mice (Fig. 5c, d).

## Discussion

We demonstrate here that the RNA-binding protein, ZFP36L1, regulates OA pathogenesis by modulating the mRNA stability of the HSP70 family members, HSPA1A and HSPA1B. We show in human and mouse chondrocytes that ZFP36L1 is upregulated by mechanical stress (i.e., DMM surgery), extracellular catabolic regulators (i.e., pro-inflammatory cytokines), and cellular catabolic mediators (i.e., HIF-2α and ZIP8). The upregulation of ZFP36L1 is necessary for the development of OA, since our results reveal that genetic ablation or knockdown of $Zfp36l1$ inhibits post-traumatic OA in mice. We further report that knockdown of ZFP36L1 inhibits OA pathogenesis by upregulating HSPA1A. This HSP70 family member protects against chondrocyte apoptosis, and its overexpression in joint tissues protects mice against OA cartilage destruction. Collectively, our results indicate that ZFP36L1 could be a therapeutic target in novel efforts to combat OA pathogenesis.

A key initial finding obtained from screening RNA-binding proteins and ZFP36 family members was the observation that ZFP36L1 is specifically upregulated in OA chondrocytes. Different ZFP36 family members play distinct cellular functions[14–20], and disruption of their specific encoding genes in mice has been reported to yield isoform-specific and non-overlapping phenotypes[14,21–23]. However, some cellular functions are regulated by multiple members of the ZFP36 family. For instance, both ZFP36L1 and ZFP36L2 promote cell quiescence[33], thymic development, and T-lymphoblastic leukemia[34]. Here, we found that the mRNA level of ZFP36L1, but not those of other ZFP36 family members, was specifically increased in chondrocytes subjected to stimulation of OA-associated catabolic signaling. This

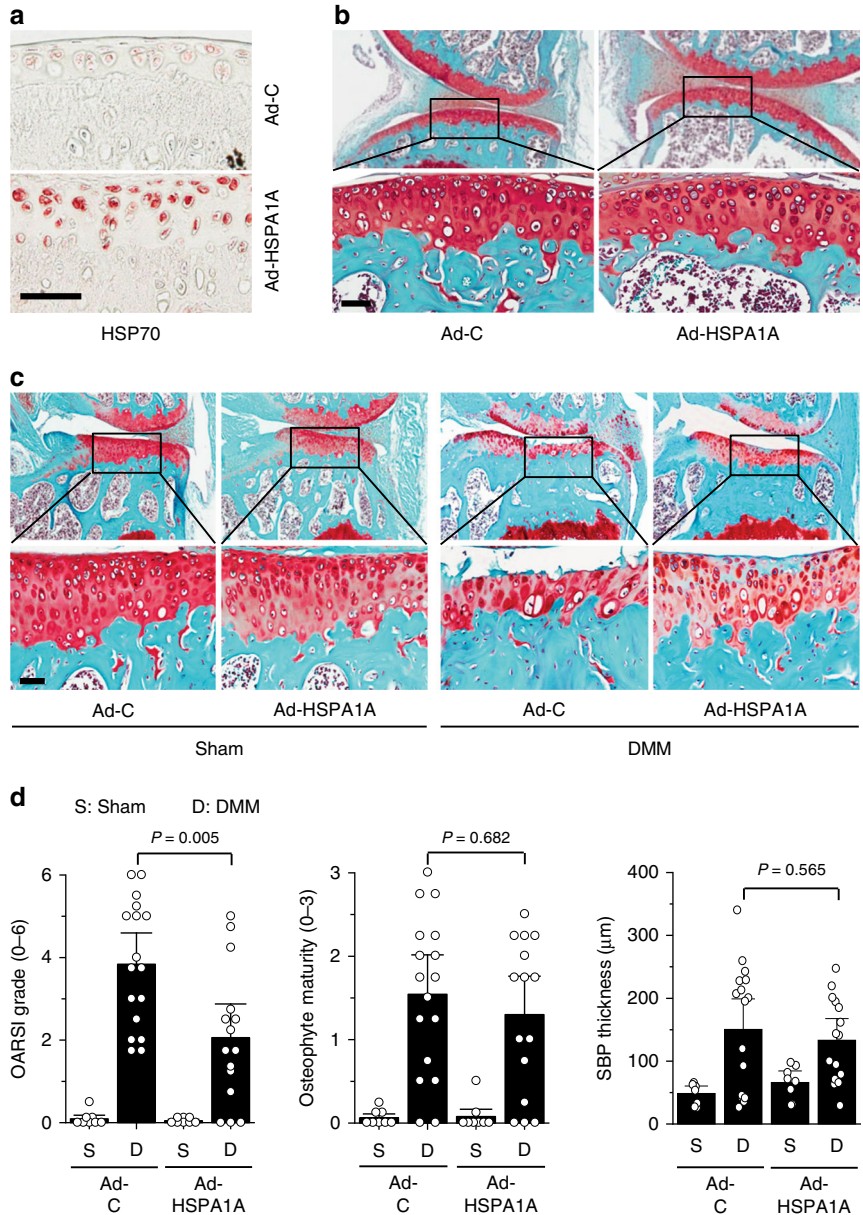

**Fig. 5** HSPA1A protects against DMM-induced cartilage destruction. **a**, **b** Representative images of HSPA1A immunostaining in cartilage (**a**) and Safranin-O staining in joint sections (**b**) of mice IA injected with Ad-C or Ad-HSPA1A ($n = 5$ mice per group) once a week for 3 weeks. **c**, **d** Representative Safranin-O staining images (**c**) and scoring of OARSI grade, osteophyte maturity, and subchondral bone plate thickness (**d**) in sham ($n = 8$ mice per group)- or DMM ($n = 17$ mice for Ad-C and 15 mice for Ad-HSPA1A)-operated mice subjected to IA injection of Ad-C as a control or Ad-HSPA1A to overexpress HSP70 in the joint tissues. Means ± 95% CI with Mann–Whitney $U$ test for OARSI grade and osteophyte maturity. Means ± s.e.m. with two-tailed $t$-test for SBP thickness. SBP subchondral bone plate. Scale bar: 50 μm

strongly suggested that ZFP36L1 could have a specific regulatory function in OA pathogenesis. The isoform-specific and redundant functions of ZFP36 family members may reflect their actions on specific mRNA targets. Indeed, while individual isoforms regulate specific target mRNAs, some mRNAs, such as those encoding TNF-α, granulocyte macrophage colony-stimulating factor (GM-CSF), and IL-3, are regulated redundantly by ZFP36, ZFP36L1, and ZFP36L2 (refs. [15–17]).

The in vivo relevance of the upregulation of ZFP36L1 in OA chondrocytes was clearly demonstrated by our observation that the disruption or downregulation of *Zfp36l1* significantly abrogated DMM-induced OA pathogenesis in mice. We used *Zfp36l1*[+/−] mice for our experimental OA study because the corresponding homozygous mice (*Zfp36l1*[−/−]) are not viable[22]. We

found that deletion of one allele of *Zfp36l1* was sufficient to inhibit the OA pathogenesis induced by DMM surgery. Similarly, our previous work demonstrated that mice heterozygous for other deletions, including those of *Epas1* (encoding HIF-2α)[11], *Slc39a8* (ZIP8)[12], and *Esrrg* (ERRγ)[24], exhibit significant reductions in DMM-induced OA pathogenesis. These molecules are essential catabolic regulators of OA pathogenesis, and their homozygous deletions are lethal. The use of a chondrocyte-specific conditional KO (cKO) system, instead of whole-body heterozygous KO, might provide valuable insights into the role of ZFP36L1 in chondrocytes during OA cartilage destruction. Because we focused mainly on a role for ZFP36L1 in chondrocytes, the limitation of current studies is the lack of chondrocyte-specific cKO system. However, OA is a whole-joint disease that involves

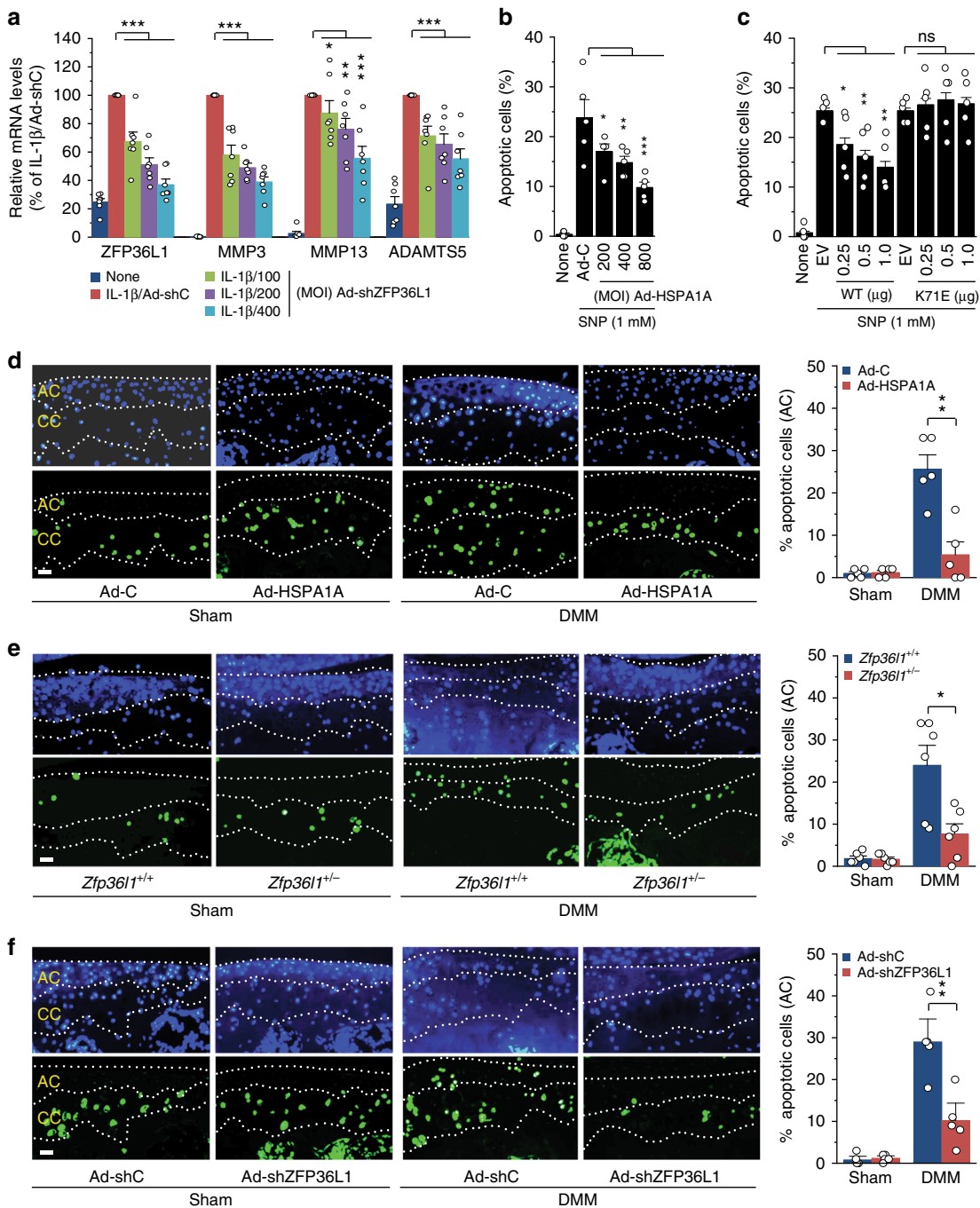

**Fig. 6** HSPA1A inhibits chondrocyte apoptosis. **a** qRT-PCR analysis ($n = 7$) of matrix-degrading enzymes in chondrocytes treated with IL-1β and infected with 400 MOI of control virus or the indicated MOIs of Ad-shZFP36L1. **b**, **c** Chondrocytes were infected with Ad-C or Ad-HSPA1A (**b**) or transfected with empty vector (EV, 1 μg) or vectors encoding WT-HSPA1A or K71E-HSPA1A (**c**) and left untreated or exposed to the NO donor, SNP, for 6 h. Apoptotic chondrocytes were identified and quantified by TUNEL staining ($n = 5$). **d**–**f** Detection and quantitation of apoptotic chondrocytes in calcified cartilage (CC) and articular cartilage (AC) of sham- or DMM-operated mice subjected to IA injection with Ad-C or Ad-HSPA1A ($n = 5$ mice per group) (**d**), sham- or DMM-operated $Zfp36l1^{+/−}$ and WT mice (**e**; $n = 6$ mice per group), or sham- or DMM-operated mice subjected to IA injection with Ad-shC or Ad-shZFP36L1 (**f**; $n = 5$ mice per group). Means ± s.e.m. with one-way ANOVA (**a**–**c**) and two-tailed t-test (**d**–**f**). *$P < 0.05$, **$P < 0.005$, and ***$P < 0.0005$; ns, not significant. Scale bar: 50 μm

multiple cell types and multiple pathological changes in all joint tissues[2,3]. Therefore, whole-body deletion and whole-joint knockdown of a target gene may be alternative strategies than its cell-type-specific alteration when seeking to examine OA as a whole-joint disease. One interesting finding of the current study is that although deletion or knockdown of $Zfp36l1$ inhibited OA pathogenesis, ZFP36L1 overexpression in joint tissues was not

sufficient to cause changes in cartilage structure or synovial inflammation. Indeed, we did not detect any marked modulation of HSPA1A mRNA levels in chondrocytes overexpressing ZFP36L1. This might reflect that HSPA1A is expressed at a low level in the absence of extracellular stress.

As ZFP36L1 regulates cellular functions by enhancing the decay of target mRNAs[14–17], we examined whether direct effector

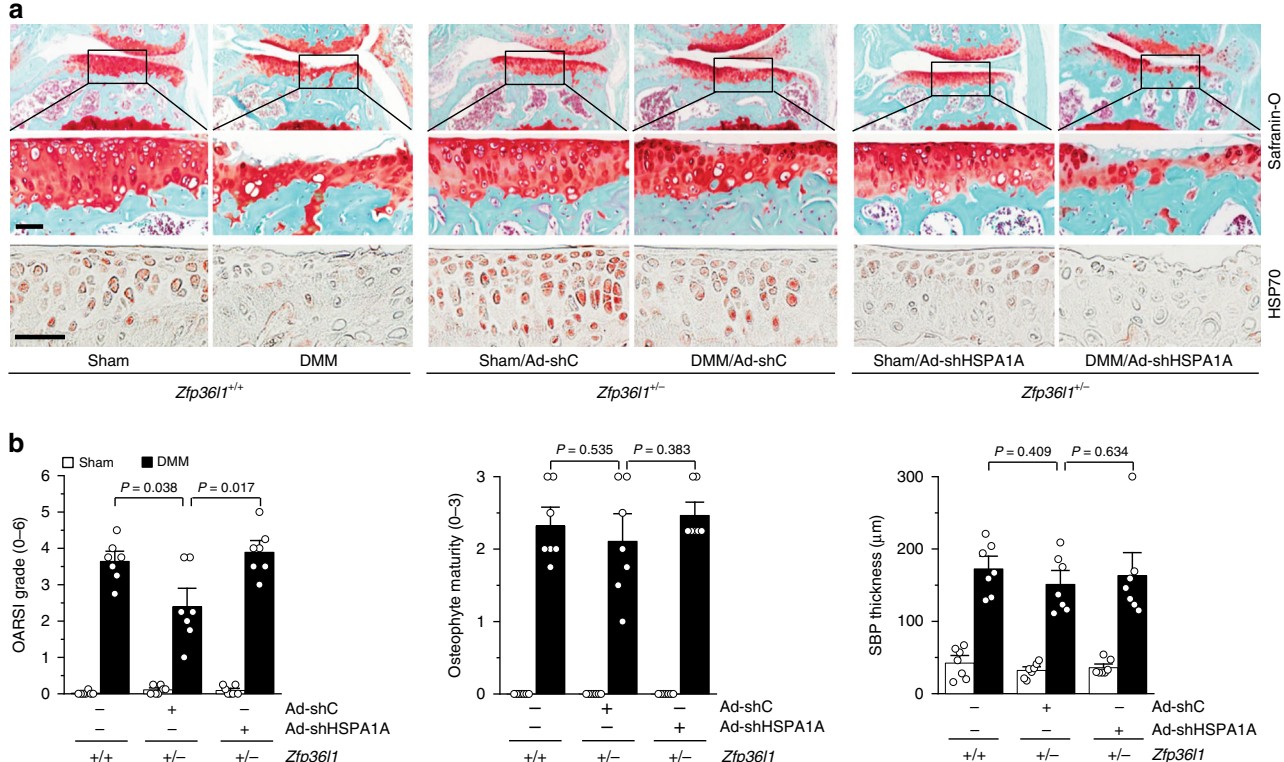

**Fig. 7** Knockdown of HSPA1A abrogates the inhibition of OA pathogenesis seen in *Zfp36l1*[+/−] mice. Sham- or DMM-operated WT and *Zfp36l1*[+/−] mice were IA injected with Ad-shC or Ad-shHSPA1A (*n* = 7 mice per group) and sacrificed at 8 weeks after the operation. **a** Representative images of Safranin-O staining and immunostaining of HSP70. **b** Quantitation of OA parameters (OARSI grade, osteophyte maturity, and subchondral bone plate thickness). Means ± 95% CI with Mann–Whitney *U* test for OARSI grade and osteophyte maturity. Means ± s.e.m. with two-tailed *t*-test for SBP thickness. Scale bar: 50 μm

molecules of OA cartilage destruction, such as matrix-degrading enzymes and cartilage ECM molecules, are targeted by ZFP36L1 in chondrocytes. We found that although the examined matrix-degrading enzymes (MMP3, MMP13, and ADAMTS5), ECM molecules (type II collagen and aggrecan), and SOX9 all contained AREs in their 3′-UTRs, they are not direct targets of ZFP36L1. Interestingly, knockdown of ZFP36L1 inhibited the IL-1β-induced upregulation of MMP3, MMP13, and ADAMTS5. However, knockdown of ZFP36L1 alone in the absence of IL-1β did not affect the mRNA levels of these enzymes, indicating that ZFP36L1 indirectly modulates the IL-1β-induced increase of these matrix-degrading enzymes. Recently, McDermott et al.[35] demonstrated that the siRNA-mediated downregulation of ZFP36 in human articular chondrocytes and SW1353 chondrosarcoma cells regulated the mRNA levels of SOX9 and aggrecan, whereas that of MMP13 was regulated by the RNA-binding protein, human antigen R (HuR). In the present study, we found that OA-associated catabolic signaling did not modulate the expression of ZFP36 in primary-culture mouse articular chondrocytes. Therefore, future studies are needed to assess the in vivo relevance of the ability of ZFP36 to regulate SOX9 and aggrecan. The ZFP36L1 targets that have been identified in various cell types include TNF-α, GM-CSF, IL-3, vascular endothelial growth factor (VEGF), cellular inhibitor of apoptosis protein-2 (c-IAP2), steroidogenic acute regulatory protein, low density lipoprotein receptor (LDLR), Notch1, Stat5b, Bcl2, Cdk5, etc.[17,31,36–40]. However, our microarray analysis indicated that the mRNA levels of the ZFP36L1 targets reported in other cell types were not markedly modulated by ZFP36L1 knockdown in primary-culture chondrocytes. Instead, we unexpectedly found that the HSP70 family members, HSPA1A and HSPA1B, are direct targets of

ZFP36L1 in chondrocytes. Of the mRNAs altered by the Ad-shZFP36L1-mediated knockdown of ZFP36L1 in chondrocytes, the mRNAs encoding HSPA1A and HSPA1B showed the most marked increases. Furthermore, our RIP and mRNA decay assays clearly indicated that the mRNA encoding HSPA1A is a direct target of ZFP36L1. Although we focused on HSPA1A in the present work, our results indicated that HSPA1B mRNA would also be a target of a ZFP36L1. The 3′-UTRs of HSPA1A and HSPA1B both contain several AREs, and are likely to undergo similar regulation by ZFP36L1. Additionally, the sequences of HSPA1A and HSPA1B differ by only 10 nucleotides and one amino acid[29].

Consistent with the ZFP36L1 knockdown-induced inhibition of DMM-induced OA pathogenesis, overexpression of the ZFP36L1 target, HSPA1A, in mouse joint tissues also inhibited DMM-induced cartilage destruction. Moreover, knockdown of HSPA1A in *Zfp36l1*[+/−] mice abolished the inhibitory effects of *Zfp36l1* KO on cartilage destruction in DMM-operated mice. We additionally found that HSPA1A inhibited the NO-induced apoptosis of primary-culture chondrocytes and decreased DMM-induced chondrocyte apoptosis in vivo. Our results are fully consistent with the known functions of HSP70 family members in chondrocytes and cartilage. For instance, studies have shown that the protein levels of HSP70 are elevated in chondrocytes of human and mouse OA cartilage[41,42], and that various anti-apoptotic agents, including glutamine, hyperbaric oxygen, and *N*-{(1Z)-5,6-dimethyl-3-oxo-8-[(4,4,5,5,5-penta-fluoropentyl)thio]-2,3-dihydro-1*H*-imidazo[5,1-c][1,4]thiazin-1-ylidene}-4-methylbenzenesulfonamide (ITZ-1), inhibit NO-induced chondrocyte apoptosis via overexpression of the HSP70 protein[43–45]. Indeed, experimentally overexpressed HSP70 was

found to prevent NO-induced apoptosis in rabbit articular chondrocytes[46]. Our results are consistent with those of a recent report by Grossin et al., in which gene transfer of HSP70 in rat chondrocytes was found to confer cytoprotection in vitro and during experimental OA[47]. Although accumulating evidence indicates that upregulated HSP70 in chondrocytes exerts protective effects, little was previously known about the upstream signaling molecules responsible for regulating HSP70 expression. Therefore, our results provide novel insight into the regulatory mechanisms through which HSP70 family members are upregulated in chondrocytes.

We found that genetic ablation or knockdown of ZFP36L1 abrogated all examined manifestations of DMM-induced OA (cartilage erosion, osteophyte formation, and subchondral bone plate thickening). However, overexpression or knockdown of HSPA1A appeared to modulate only DMM-induced cartilage erosion, as no change was seen in osteophyte formation or subchondral bone plate thickening. OA is well recognized as a whole-joint disease that occurs in all tissues of the affected joint[2,3]. However, the contributions and relationships of the events seen during OA pathogenesis are not yet clear. For instance, it has long been debated whether subchondral bone sclerosis acts as a driving force for cartilage breakdown or arises as a consequence of articular cartilage catabolism[2,3]. As summarized by Loeser et al.[2], it is becoming clear that different mediators regulate each event. For instance, osteophyte formation is regulated by transforming growth factor (TGF)-β and bone morphogenetic protein (BMP)-2; subchondral bone sclerosis is mediated by Wnt/β-catenin, sclerostin, BMPs, and insulin like growth factor (IGF)-1; and cartilage erosion is mediated by upregulation of matrix-degrading enzymes, downregulation of ECM molecules, and apoptosis of chondrocytes. Therefore, ZFP36L1 appears to regulate multiple factors that are responsible for mediating cartilage erosion, osteophyte formation, and subchondral bone sclerosis, whereas downstream targets of HSPA1A regulate mediators of cartilage destruction.

In summary, we herein demonstrate that OA involves the specific upregulation of the RNA-binding protein, ZFP36L1, which regulates the HSPA1A mRNA as a direct target. HSPA1A protects against OA by inhibiting chondrocyte apoptosis. Additionally, either downregulation of ZFP36L1 or overexpression of HSPA1A in joint tissues is sufficient to inhibit experimental OA in mice. Our results collectively suggest that the ZFP36L1 could be an effective therapeutic target in the fight against OA, as overexpression of the ZFP36L1 target, HSPA1A, is sufficient to inhibit post-traumatic OA in mice.

## Methods

**Human OA cartilage and experimental OA in mice.** Human OA cartilage was sourced from individuals undergoing arthroplasty[11,12,24–26]. The Institutional Review Board of Wonkwang University Hospital approved the use of these materials, and all participants provided written informed consent before the operative procedure. We have complied with all relevant ethical regulations with approved study protocols. C57BL/6J mice were used for the experimental OA studies. Zfp36l1[+/−] mice (8-bp insertion/42-bp deletion in exon 2) were generated by ToolGen, Inc. All experiments were approved by the Animal Care and Use Committee of Gwangju Institute of Science and Technology. We have complied with all relevant ethical regulations with approved study protocols. Experimental OA was induced in 12-week-old male mice by DMM surgery or by IA injection (once weekly for 3 weeks) of adenovirus ($1 \times 10^9$ plaque-forming units [PFUs] in a total volume of 10 μl) expressing ZFP36L1 (Ad-ZFP36L1), HSP70 (Ad-HSPA1A), ZIP8 (Ad-ZIP8), or HIF-2α (Ad-HIF-2α)[11,12,24–26]. Mice were sacrificed 8 weeks after DMM surgery or 3 or 8 weeks after the first IA injection, and subjected to histological and biochemical analyses.

**Histology and immunohistochemistry.** Human OA cartilage was frozen, sectioned at a thickness of 5 μm, and fixed in paraformaldehyde. Sulfate proteoglycan was detected with Alcian blue staining. Mouse knee joints were fixed in 4% paraformaldehyde, decalcified in 0.5 M EDTA, and embedded in paraffin. The paraffin

blocks were sectioned at a thickness of 5 μm and the sections were deparaffinized in xylene, hydrated with graded ethanol, and stained with Safranin-O. Cartilage destruction was scored by three observers under blinded conditions using the OARSI scoring system (grade 0–6)[48]. The OARSI scores are presented as the mean of the maximum score in each mouse, and each representative Safranin O-stained image was selected from the most advanced lesion among serial sections. Synovitis (grade 0–3) was determined by Safranin-O and hematoxylin staining[24–26]. Osteophyte formation was identified by Safranin-O staining, and osteophyte size was measured with an Aperio Image Scope V12 (Leica Biosystems)[24–26]. Subchondral bone sclerosis was determined by measuring the thickness of the subchondral bone plate[24–26]. ZFP36L1, HSP70, MMP3, and MMP13 were detected by immunohistochemical staining of human and mouse cartilage sections, as performed using following antibodies: rabbit polyclonal antibody against ZFP36L1 (1:100 dilution, ab209419; Abcam); rabbit polyclonal antibody (1:1000 dilution, AF1663; R&D systems) or mouse monoclonal antibody (3.5 μg ml$^{-1}$, ab5442; Abcam) against HSP70; rabbit polyclonal antibody (4 μg ml$^{-1}$, ab52915; Abcam) against MMP3; and rabbit polyclonal antibody (1:100 dilution, ab51072; Abcam) against MMP13.

**Skeletal staining.** Skeletons of whole-mouse embryos were stained with Alcian blue and Alizarin red[24–26]. Briefly, whole embryos were skinned, eviscerated, fixed with 95% ethanol for 4 days, and immersed in acetone for 3 days. Samples were stained in a freshly prepared staining solution composed of 1 volume of 0.3% Alcian blue 8GX in 70% ethanol, 1 volume of 0.1% Alizarin red S in 95% ethanol, 1 volume of 100% acetic acid, and 17 volumes of 100% ethanol. The samples were sequentially destained with 1% KOH for up to 48 h and with 20% glycerol containing 1% KOH for 14 days. Images were acquired using a Zeiss SteREO Discovery V12 microscope.

**Primary culture of articular chondrocytes.** Chondrocytes were isolated from the femoral condyles and tibial plateaus of postnatal day 5 WT or Zfp36l1[+/−] mice by digestion of cartilage tissue with 0.2% collagenase[49]. The cells were maintained as a monolayer in Dulbecco's modified Eagle's medium (DMEM; Gibco) supplemented with 10% fetal bovine serum and antibiotics (penicillin G and streptomycin). Passage 0 cell were used for analyses unless otherwise indicated.

**Infection and IA injection in mice.** The adenoviruses Ad-C (1060), Ad-ZFP36L1 (ADV-276800), Ad-HSPA1A (ADV-277566), and Ad-ZIP8 (ADV-272407) were purchased from Vector Biolabs. The adenovirus-expressing mouse HIF-2α was produced by Newgex. The control shRNA (1122N), Ad-shHSPA1A (shADV-277566), and Ad-shZFP36L1 (shADV-276800) were purchased from Vector Biolabs. Mouse articular chondrocytes were cultured for 2 days, infected with the indicated MOIs of adenovirus for 2 h, and cultured in the presence or absence of pharmacological agents for an additional 24 h. Adenovirus ($1 \times 10^9$ PFUs in a total volume of 10 μl) was injected into the knee joints of mice once per week for 3 weeks[11,12,24–26]. Mice were sacrificed 3 or 8 weeks after the first injection of adenovirus.

**Western blotting.** Total cell lysates were prepared in lysis buffer (150 mM NaCl, 1% NP-40, 50 mM Tris, 0.2% sodium dodecyl sulfate [SDS], 5 mM NaF) and used to detect ZFP36L1 or HSP70. All lysis buffers contained a cocktail of protease inhibitors and phosphatase inhibitors (Roche). The following antibodies were used for western blotting: rabbit polyclonal anti-ZFP36L1 (10 μg ml$^{-1}$, SC-134091; Santa Cruz Biotech.); rabbit polyclonal anti-HSP70 (1:1000 dilution, #4872; Cell Signaling); anti-MMP3 (clone EP1186Y, 1 μg ml$^{-1}$, ab52915; Abcam); anti-MMP13 (clone EP1263Y, 1:1000 dilution, ab51072; Abcam); anti-GFP (1:4000 dilution, ab290; Abcam); anti-lamin B (1:2000 dilution, SC-6216; Santa Cruz Biotech.); and anti-ERK (1:2000 dilution, 610408; BD Biosciences).

**RT-PCR and qRT-PCR.** Total RNA was extracted from primary-culture chondrocytes using the TRI reagent (Molecular Research Center, Inc.). Total RNA was reverse transcribed, and the resulting cDNA was PCR-amplified using the PCR primers and experimental conditions summarized in Supplementary Table 4. qRT-PCR reactions were performed using an iCycler thermal cycler (Bio-Rad) and SYBR premixExTaq. For each target gene, the transcript levels were normalized with respect to that of β-actin and expressed as a fold-change relative to the indicated control.

**mRNA decay assay.** Primary-culture chondrocytes were infected with 800 MOI of Ad-HSPA1A to overexpress HSPA1A. The cells were co-infected with 800 MOI of Ad-C or Ad-ZFP36L1 for 12 h, and exposed to actinomycin D (1 μg ml$^{-1}$) for the indicated time periods. Total RNA was reverse transcribed and HSPA1A mRNA levels were quantified by qRT-PCR. The mRNA levels were normalized to those of glyceraldehyde-3-phosphate dehydrogenase (GAPDH).

**RIP assay.** The RNA-binding protein immunoprecipitation assay was performed using a Magna RIP™ RNA-Binding Protein Immunoprecipitation Kit (Millipore) according to the manufacturer's instructions. Briefly, 70% confluent mouse

articular chondrocytes were infected with Ad-ZFP36L1 and Ad-HSPA1A at the indicated MOIs for 24 h. Cell lysates were immunoprecipitated with anti-ZFP36L1 (3 μg, 101AP; FabGennix International) or control IgG (1 μg, CS200621; EMD Millipore Corp.) antibody. RNA was extracted from immobilized magnetic bead-bound complexes. The immunoprecipitated RNA was reverse transcribed, and the resulting cDNA was PCR-amplified. qRT-PCR reactions were performed using an iCycler thermal cycler (Bio-Rad) and SYBR premixExTaq (TaKaRa Bio). The primers for the RIP assay were designed to amplify three different ARE-containing regions of the Hspa1a 3′-UTR region (Supplementary Table 4). For each target gene, the transcript levels were normalized that of GAPDH and expressed as a fold-change relative to the indicated control.

**Microarray analyses**. Microarray analysis of chondrocytes stimulated by IL-1β treatment or overexpression of HIF-2α or ZIP8 was performed as described previously[11,12,24–26]. We also performed microarray analysis in chondrocytes infected with Ad-ZFP36L1 or Ad-shZFP36L1. Briefly, total RNA was extracted from mouse articular chondrocytes using a Purelink RNA mini kit (Ambion). The concentration, purity, and integrity of the extracted RNA were determined by NanoDrop 2000 spectrophotometry (Thermo Scientific). RNA from mouse chondrocytes was analyzed using Affymetrix Gene Chip arrays (Affymetrix GeneChip Mouse Gene 2.0 ST Array) in accordance with the Affymetrix protocol (Macrogen Inc.). The probe signals in the raw data were normalized with respect to the RMA (Robust Multi-array Average) for each separate data set (infection of Ad-C, Ad-ZFP36L1 or Ad-shZFP36L1). Normalization was performed using R v.3.3.2 with Affy package v.1.52.0 (Affymetrix Inc.). To identify DEGs (differentially expressed genes), we performed the Student's t-test followed by the Benjamini–Hochberg multiple hypothesis test for each group using Python v.3.4.3 and StatsModels library v.0.8.0 (Python Software Foundation). The cutoff values for DEG identification were adjusted such that the P-value was smaller than 0.05 (FDR < 0.05) and the absolute value of the Log Fold Change was greater than 1 (|LogFC| > 1).

**Apoptosis assay**. Primary-culture chondrocytes were infected with 800 MOI of Ad-C or Ad-HSPA1A for 24 h. The cells were exposed to the NO donor, SNP, to induce apoptosis[30]. Apoptotic chondrocytes were detected by TUNEL (terminal deoxynucleotidyl transferase dUTP nick-end labeling) assay using a kit from Roche Diagnostics. To examine apoptosis of chondrocytes in cartilage tissue, sham- or DMM-operated mice were IA injected with Ad-C or Ad-HSPA1A to overexpress the HSPA1A protein, and the TUNEL assay was used to detect apoptotic chondrocytes in joint sections.

**Statistical analysis**. All statistical analyses were performed using the IBM SPSS Statistics 21 software. For cell-based in vitro studies, a two-tailed Student's t-test with unequal sample sizes and variances was used for pair-wise comparisons and one-way analysis of variance (ANOVA) with post hoc tests was used for multi-comparisons. Data collected from mouse experiments were analyzed using the nonparametric Mann–Whitney U test. Each n number indicates the number of biologically independent samples, mice per group, or human specimens. Significance was accepted at the 0.05 level of probability (P < 0.05). Each bars represent s.e.m. for parametric data and the calculated 95% confidence intervals (CIs) for nonparametric data.

**Reporting Summary**. Further information on experimental design is available in the Nature Research Reporting Summary linked to this article.

## Data availability
The microarray data have been deposited to the Gene Expression Omnibus under accession codes GSE104794 (for HIF-2α), GSE104795 (for ZIP8), GSE104793 (for IL-1β), and GSE110581 (for ZFP36L1). All other data supporting the findings of this study are available within the paper and its supplementary information files.

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

## Acknowledgements

This work was supported by grants from the National Research Foundation of Korea (2016R1A3B1906090 and 2016R1A5A1007318) and the Korea Health Technology R&D Project through the Korea Health Industry Development Institute (H114C3484 and HI16C0287). Y.O.S. was supported by grants from the Basic Science Research Program through the National Research Foundation of Korea (2016R1D1A1B03930327).

## Author contributions

Y.-O.S. and H.-E.K. performed study design, data acquisition, analysis, and interpretation, and manuscript preparation and approval; W.-S.C. performed data acquisition, analysis, and interpretation, and manuscript approval; C.-H.C. provided and evaluated human joint samples; J.-S.C. (jschun@gist.ac.kr) takes responsibility for the integrity of this work.

## Additional information

**Competing interests:** The authors declare no competing interests.

