## [Peer Review File · Nature Communications]

Reviewers' Comments:

Reviewer #1:

Remarks to the Author:

GENERAL COMMENTS: This manuscript reports findings that are nicely follow up the previous reports of this group on cellular metabolic pathways that play roles in cartilage homeostasis and/or dysfunction. Of interest is that ZFP3611, among the RNA binding proteins of the ZFP36 family, appears to have a specific role in OA chondrocytes. The mechanism that the authors seem to be proposing is that upregulation of ZFP3611 in OA cartilage could promote OA by upregulating apoptosis in cartilage through down-regulating HSPA1A, a chondroprotective intracellular protein. The authors also showed that ZFP3611 promotes cartilage destruction possibly by indirect effects on IL-1-induced matrix-degrading proteins, but its ablation also attenuates osteophyte formation and subchondral bone plate thickness, which are not regulated by HSPA1A. Thus, these studies define a novel mechanism involving specific regulation by an RNA binding protein, but more work may be needed to determine how to precisely regulate this pathway for therapeutic intervention.

SPECIFIC COMMENTS:

1. Introduction: It is unclear why the authors decided to specifically assess RNA binding proteins by microarray analysis in the context of OA pathogenesis. Is this because the class of genes appeared in their previous studies (e.g., in relation to ZIP8), or are there previously published studies indicating that RNA binding proteins may play a role in OA?

2. In the concluding two paragraphs of the Discussion, it is unclear whether both ZFP3611 and HSPA1A should be considered the therapeutic targets. Since HSPA1A appears to play a protective role, it might be a therapeutic molecule itself, whereas ZFP3611 could be a target because its deletion abrogates OA in mice. Several statements are confusing in that regard, for example, where it is stated that ZFP3611 regulates mediators responsible for cartilage destruction, etc., and downstream targets of HSPA1A regulate mediators of cartilage destruction. Please consider rewriting these concluding paragraphs to make them less confusing.

3. Methods, Page 15, lines 410/411: Subchondral bone sclerosis is a result of complex changes in tissue structure, organization, and mineralization. What is really being measured is subchondral bone plate thickness, which is correctly used in the labels on the graphs, so this terminology should be used in describing the results for Figure 2 (page 6, line 149), Figure 3 (page 7, lines 167 and 172), and Figure 5 (page 9, line 217) and in the Discussion (line 359, 361, etc.).

MINOR:

Figure 6: correct "ADATS5" to "ADAMTS5".

Reviewer #2:

Remarks to the Author:

An interesting study implicating the RNA-binding protein ZFP3611 as a mediator of processes involved in the development of osteoarthritis. This is timely as there is increasing interest in the role of post-transcriptional and protein translational control as mediators of this disease. The study uses a number of in vivo and in vitro approaches to come to its conclusion. I do feel there are some issues that need to be addressed however.

1. Primary chondrocytes de-differentiate quickly in culture and so detail needs to be provided regarding the human and murine primary chondrocyte work. Specifically, this should include the time in culture of the cells before they were used (e.g. passage number) and the quantitative expression of marker genes in these cells. Building on this, in conditions used to model osteoarthritis responses (e.g. IL-1 stimulation, or adenoviral overexpression of Hif2a and Zip8)

both human and murine systems need to be properly characterised by determining positive regulation of genes such as MMP13 at a quantitative level or by examining increased enzyme activity.

2. Zfp3611 protein levels are shown to be higher in both human OA cartilage and in mouse OA model systems using immunohistochemistry. For the human tissue, shown in figure 1c, there is now description of whether the micrographs shown represent anatomically equivalent regions of the joint surface (e.g. location on surface and depth of tissue). This is important as different locations might see different levels of gene/protein expression. Full depth sections from equivalent joint regions should be presented with better histological characterisation of proteoglycan loss and tissue fibrillation. In the same figure, the mouse OA model shows more clearly an increase in Zfp3611 in the cartilage from DMM-treated mice or those injected with Hif2a or Zip8 adenovirus. In the controls in these cases there appears to be very little ZFP3611. However, in the KO mouse analysis shown in supplementary figure 2c there is a lot of ZFP3611 visible in the cartilage of control mice (I'm assuming they are controls although I think they are incorrectly labelled '+/-' in supplementary figure 2c). On the surface there appears to be a mixed message given by these different figure with low expression being shown in control cartilage in figure 1 but high expression portrayed in supplementary figure 2C.

3. The use of the knockout mouse model does not appear to me to be totally conclusive. These mice are stated as having no developmental defects, but the evidence for this is limited to the skeletal staining in supplementary figure 2D. This analysis needs to be far more rigorous. Measurement of the length of skeletal elements during development and post-nataly as well as histological confirmation that development remains the same is required. It is odd that the authors have not decided to use a conditional knockout system to specifically determine the consequence of loss of ZFP3611 in cartilage. This could complement the heterozygous null mouse mouse and determine whether the role of the gene in OA is due to its function in cartilage or in other tissues.

4. The use of adenoviruses to overexpress the genes of interest (ZFP3611 and HSPA1) or in some cases to express shRNAs targeting these genes is used a great deal in this study and appears to lead to strong infection and expression in the articular chondrocytes. There is very little focus however on the degree to which other joint tissues are being infected and thus expressing these factors. Evidence for modulation of ZFP3611 and HSPA1 should be measured in the synovium, cruciate ligament and meniscus as well.

5. Throughout the manuscript there is inconsistency in the gene expression analysis of the in vitro experiments regarding the use of end point or quantitative real time PCR. The latter should be used in all cases. Conclusions such as those for figure 2C where it is stated that "ZFP36L1 overexpression in primary-culture chondrocytes did not affect the expression levels of MMP3, MMP13, and ADAMTS5 which play crucial roles in OA cartilage destruction, nor did it affect cartilage ECM molecules, such as type II collagen and aggrecan, or the regulatory transcription factor, SOX9" cannot be justified by the use of end point PCR alone.

6. Is there a fundamental reason why Hif2a and Zip8 adenoviruses need to be respectively called Ad-Epas1 and Ad-Slc3908. Surely names like "Ad-Hif2a" and "Ad-Zip8" would result in far less confusion for the reader?

7. The authors focussed on the expression of HSPA1A as there were only 6 nucleotides difference between it and the HSPA1B gene. Is this within the whole mRNA (including 3-UTR) or just the coding region? If the former, then wouldn't study of the other gene also be useful as the two may exhibit distinct regulatory differences through different 3-UTR elements.

8. Figure 4C HSPA1A adenovirus in DMM model. The image for the DMM + HSPA1-Ad does not illustrate the data very well as there appears to be a significant reduction in subchondral bone sclerosis compared to the DMM Ad-C treatment, even though this parameter is not found to be

significant in the scoring.

Reviewer #3:

Remarks to the Author:

The manuscript finds a role for the RNA binding protein ZFP36L1 and the chaperone HSP70 (HSPA1A) in osteoarthritis. The authors use primary mouse chondrocytes, and an in vivo mouse model of surgical disruption of the knee joint (DMM). Treatments (eg. IL-1b) that promote breakdown of cartilage upregulated ZFP36L1 in chondrocytes, human patient samples and the mouse model. While ZFP36L1 overexpression did not cause any pathology, depletion in the ZFP36L1 +/- mouse or by knockdown, partially relieved the disease phenotype. In chondrocytes, HSPA1A and HSPA1B transcripts were the most highly upregulated upon ZFP36L1 knockdown. ZFP36L1 associated with one region of ARE's in the HSPA1A 3'-UTR, and overexpression of ZFP36L1 increased the turnover of HSPA1A transcript. In the DMM model, overexpression of HSPA1A in the joint partially relieved the loss of cartilage, although other phenotypes were not affected. HSPA1A also prevented apoptosis of chondrocytes in articular cartilage after DMM surgery, and in cultures of chondrocytes. It is proposed that ZFP36L1 regulation of HSPA1A may be a therapeutic target.

The findings are interesting, and reveal an unexpected mode of HSPA1A regulation. A role for ZFP36L1 has not previously been known in chondrocytes and osteoarthritis, and the results in the heterozygous mouse show its importance for the disease. HSPA1A is well established to protect against apoptosis under various conditions. Chondrocyte apoptosis is part of the osteoarthritis process and the manuscript now suggests a mechanism that regulates it.

However, the authors do not take the last step and present a full disease mechanism, that high ZFP36L1 expression in osteoarthritis causes suppression of HSPA1A, allowing apoptosis to proceed. Overexpression of ZFP36L1 in healthy mice did not have much effect, the explanation put forward is that HSPA1A levels are low in the non-stressed condition.

Recommendations:

1. Show the levels of HSPA1A protein expression in the DMM tissue, for the wild-type and ZFP36L1 +/- mouse, or with its knockdown. The data showing ZFP36L1 regulation of HSPA1A are only in chondrocytes in Fig. 4. Supplemental Fig. 4 shows some expression of HSPA1A in response to HIF-2a and ZIP8, suggesting there is some expression in the disease state with wild-type ZFP36L1 levels.
2. In Fig. 4b and c in chondrocytes, show the protein levels of HSPA1A by Western blot. A 10-fold change in mRNA does not always mean that much difference in protein expression. It would be preferable to show ZFP36L1 levels by Western blot too. The effect on HSPA1A protein should also be addressed in a ZFP36L1 co-expression experiment like in Fig. 4e.
3. Show articular chondrocyte apoptosis is decreased in the ZFP36L1 +/- mouse and with its knockdown, comparable to Fig. 6d.
4. Knockdown HSPA1A in the ZFP36L1 +/- mouse to show that it restores the disease phenotype after DMM surgery.
5. A catalytically inactive mutant of HSPA1A (e.g. K71E) is a better control for the overexpression experiments in Fig. 6.

Point-to-point response to reviewers' comments

Reviewers' comments:

Reviewer #1

GENERAL COMMENTS: This manuscript reports findings that are nicely follow up the previous reports of this group on cellular metabolic pathways that play roles in cartilage homeostasis and/or dysfunction. Of interest is that ZFP36L1, among the RNA binding proteins of the ZFP36 family, appears to have a specific role in OA chondrocytes. The mechanism that the authors seem to be proposing is that upregulation of ZFP36L1 in OA cartilage could promote OA by upregulating apoptosis in cartilage through down-regulating HSPA1A, a chondroprotective intracellular protein. The authors also showed that ZFP36L1 promotes cartilage destruction possibly by indirect effects on IL-1 β -induced matrix-degrading proteins, but its ablation also attenuates osteophyte formation and subchondral bone plate thickness, which are not regulated by HSPA1A. Thus, these studies define a novel mechanism involving specific regulation by an RNA binding protein, but more work may be needed to determine how to precisely regulate this pathway for therapeutic intervention.

SPECIFIC COMMENTS:

1. Introduction: It is unclear why the authors decided to specifically assess RNA binding proteins by microarray analysis in the context of OA pathogenesis. Is this because the class of genes appeared in their previous studies (e.g., in relation to ZIP8), or are there previously published studies indicating that RNA binding proteins may play a role in OA?

2. In the concluding two paragraphs of the Discussion, it is unclear whether both ZFP36L1 and HSPA1A should be considered the therapeutic targets. Since HSPA1A appears to play a protective role, it might be a therapeutic molecule itself, whereas ZFP36L1 could be a target because its deletion abrogates OA in mice. Several statements are confusing in that regard, for example, where it is stated that ZFP36L1 regulates mediators responsible for cartilage destruction, etc., and downstream targets of HSPA1A regulate mediators of cartilage destruction. Please consider rewriting these concluding paragraphs to make them less confusing.

3. Methods, Page 15, lines 410/411: Subchondral bone sclerosis is a result of complex changes in tissue structure, organization, and mineralization. What is really being measured is subchondral bone plate thickness, which is correctly used in the labels on the graphs, so this terminology should be used in describing the results for Figure 2 (page 6, line 149), Figure 3 (page 7, lines 167 and 172), and Figure 5 (page 9, line 217) and in the Discussion (line 359, 361, etc.).

MINOR: Figure 6: correct "ADATS5" to "ADAMTS5".

Reviewer #2

An interesting study implicating the RNA-binding protein ZFP36L1 as a mediator of processes involved in the development of osteoarthritis. This is timely as there is increasing interest in the role of post-transcriptional and protein translational control as mediators of this disease. The study uses a number of in vivo and in vitro approaches to come to its conclusion. I do feel there are some issues that need to be addressed however.

1. Primary chondrocytes de-differentiate quickly in culture and so detail needs to be provided regarding the human and murine primary chondrocyte work. Specifically, this should include the time in culture of the cells before they were used (e.g. passage number) and the quantitative expression of marker genes in these cells. Building on this, in conditions used to model osteoarthritis responses (e.g. IL-1 β stimulation, or adenoviral overexpression of HIF-2 α and ZIP8) both human and murine systems need to be properly characterized by determining positive regulation of genes such as MMP13 at a quantitative level or by examining increased enzyme activity.

2. ZFP36L1 protein levels are shown to be higher in both human OA cartilage and in mouse OA model systems using immunohistochemistry. For the human tissue, shown in figure 1c, there is now description of whether the micrographs shown represent anatomically equivalent regions of the joint surface (e.g. location on surface and depth of tissue). This is important as different locations might see different levels of

gene/protein expression. Full depth sections from equivalent joint regions should be presented with better histological characterization of proteoglycan loss and tissue fibrillation. In the same figure, the mouse OA model shows more clearly an increase in ZFP36L1 in the cartilage from DMM-treated mice or those injected with HIF-2 α or ZIP8 adenovirus. In the controls in these cases there appears to be very little ZFP36L1. However, in the KO mouse analysis shown in supplementary figure 2c there is a lot of ZFP36L1 visible in the cartilage of control mice (I'm assuming they are controls although I think they are incorrectly labelled '+/-' in supplementary figure 2c). On the surface there appears to be a mixed message given by these different figure with low expression being shown in control cartilage in figure 1 but high expression portrayed in supplementary figure 2C.

3. The use of the knockout mouse model does not appear to me to be totally conclusive. These mice are stated as having no developmental defects, but the evidence for this is limited to the skeletal staining in supplementary figure 2D. This analysis needs to be far more rigorous. Measurement of the length of skeletal elements during development and post-natally as well as histological confirmation that development remains the same is required. It is odd that the authors have not decided to use a conditional knockout system to specifically determine the consequence of loss of ZFP36L1 in cartilage. This could complement the heterozygous null mouse model and determine whether the role of the gene in OA is due to its function in cartilage or in other tissues.

4. The use of adenoviruses to overexpress the genes of interest (ZFP36L1 and HSPA1A) or in some cases to express shRNAs targeting these genes is used a great deal in this study and appears to lead to strong infection and expression in the articular chondrocytes. There is very little focus however on the degree to which other joint tissues are being infected and thus expressing these factors. Evidence for modulation of ZFP36L1 and HSPA1A should be measured in the synovium, cruciate ligament and meniscus as well.

5. Throughout the manuscript there is inconsistency in the gene expression analysis of the in vitro experiments regarding the use of end point or quantitative real time PCR. The latter should be used in all cases. Conclusions such as those for figure 2C where it is stated that "ZFP36L1 overexpression in primary-culture chondrocytes did not affect the expression levels of MMP3, MMP13, and ADAMTS5 which play crucial roles in OA cartilage destruction, nor did it affect cartilage ECM molecules, such as type II collagen and aggrecan, or the regulatory transcription factor, SOX9" cannot be justified by the use of end point PCR alone.

6. Is there a fundamental reason why HIF-2 α and ZIP8 adenoviruses need to be respectively called Ad-Epas1 and Ad-Slc39a8. Surely names like "Ad-HIF-2 α " and "Ad-ZIP8" would result in far less confusion for the reader?

7. The authors focused on the expression of HSPA1A as there were only 6 nucleotides difference between it and the HSPA1B gene. Is this within the whole mRNA (including 3-UTR) or just the coding region? If the former, then wouldn't study of the other gene also be useful as the two may exhibit distinct regulatory differences through different 3-UTR elements.

8. Figure 4C HSPA1A adenovirus in DMM model. The image for the DMM + HSPA1A-Ad does not illustrate the data very well as there appears to be a significant reduction in subchondral bone sclerosis compared to the DMM Ad-C treatment, even though this parameter is not found to be significant in the scoring.

Reviewer #3

The manuscript finds a role for the RNA binding protein ZFP36L1 and the chaperone HSP70 (HSPA1A) in osteoarthritis. The authors use primary mouse chondrocytes, and an in vivo mouse model of surgical disruption of the knee joint (DMM). Treatments (eg. IL-1 β) that promote breakdown of cartilage upregulated ZFP36L1 in chondrocytes, human patient samples and the mouse model. While ZFP36L1 overexpression did not cause any pathology, depletion in the ZFP36L1^{+/-} mouse or by knockdown, partially relieved the disease phenotype. In chondrocytes, HSPA1A and HSPA1B transcripts were the most highly upregulated upon ZFP36L1 knockdown. ZFP36L1 associated with one region of ARE's in the HSPA1A 3'-UTR, and overexpression of ZFP36L1 increased the turnover of HSPA1A transcript. In the DMM model, overexpression of HSPA1A in the joint partially relieved the loss of cartilage, although other phenotypes were not affected. HSPA1A also prevented apoptosis of chondrocytes in articular cartilage after DMM surgery, and in cultures of chondrocytes. It is proposed that ZFP36L1 regulation of HSPA1A may be a therapeutic target. The findings are interesting, and reveal an unexpected mode of HSPA1A regulation. A

role for ZFP36L1 has not previously been known in chondrocytes and osteoarthritis, and the results in the heterozygous mouse show its importance for the disease. HSPA1A is well established to protect against apoptosis under various conditions. Chondrocyte apoptosis is part of the osteoarthritis process and the manuscript now suggests a mechanism that regulates it. However, the authors do not take the last step and present a full disease mechanism, that high ZFP36L1 expression in osteoarthritis causes suppression of HSPA1A, allowing apoptosis to proceed. Overexpression of ZFP36L1 in healthy mice did not have much effect, the explanation put forward is that HSPA1A levels are low in the non-stressed condition.

1. Show the levels of HSPA1A protein expression in the DMM tissue, for the wild-type and ZFP36L1^{+/-} mouse, or with its knockdown. The data showing ZFP36L1 regulation of HSPA1A are only in chondrocytes in Fig. 4. Supplemental Fig. 4 shows some expression of HSPA1A in response to HIF-2 α and ZIP8, suggesting there is some expression in the disease state with wild-type ZFP36L1 levels.

2. In Fig. 4b and c in chondrocytes, show the protein levels of HSPA1A by Western blot. A 10-fold change in mRNA does not always mean that much difference in protein expression. It would be preferable to show ZFP36L1 levels by Western blot too. The effect on HSPA1A protein should also be addressed in a ZFP36L1 co-expression experiment like in Fig. 4e.

3. Show articular chondrocyte apoptosis is decreased in the ZFP36L1^{+/-} mouse and with its knockdown, comparable to Fig. 6d.

4. Knockdown HSPA1A in the ZFP36L1^{+/-} mouse to show that it restores the disease phenotype after DMM surgery.

5. A catalytically inactive mutant of HSPA1A (e.g. K71E) is a better control for the overexpression experiments in Fig. 6.

Summary of changes we have made to individual figures.

We originally submitted our manuscript with six main figures, six supplemental figures, and three supplementary tables. Because of the additional experiments performed during the revision process, the revised manuscript contains seven main figures, nine supplementary figures, and four supplementary tables. We also modified the appropriate text and re-organized the sequence of the experimental data presented in the main and supplemental figures. The changes made to each figure are summarized below.

Figure 1. Original Fig. 1 with modifications and additional data (b~d) to address reviewer #2's comment-2 and -5.

Figure 2. Same as original Fig. 2.

Figure 3. Same as original Fig. 3.

Figure 4. Original Fig. 4 with additional data (b, c, e) to address reviewer #2's comment-7 and reviewer #3's comment-2.

Figure 5. Original Fig. 5 with modifications (c) to address reviewer #2's comment-8.

Figure 6. Original Fig. 6 with additional data (c, e, f) to address reviewer #3's comment-3 and -5.

Figure 7. Additional data to address reviewer #3's comment-4.

Figure S1. Original Fig. 1b (b) and additional data (a,c) to address reviewer #2's comment-1.

Figure S2. Additional data (a, b) to address reviewer #2's comment-4 and -5.

Figure S3. Same as original Fig. S1.

Figure S4. Original Fig. S2 with additional data to address reviewer #2's comment-2 (c) and -3 (e, f).

Figure S5. Same as original Fig. S3.

Figure S6. Original Fig. S4 with additional data (a, c) to address reviewer #2's comment-7.

Figure S7. Additional data to address reviewer #2's comment-4 (a) and reviewer #3's comment-1 (b, c, d).

Figure S8. Original Fig. S5 with additional data (f) to address reviewer #3's comment-5.

Figure S9. Original Fig. S6 (uncropped scans of the gel images).

Table S1. Additional data related to Fig. 1a.

Table S2. Original Supplementary Table 1.

Table S3. Original Supplementary Table 2.

Table S4. Original Supplementary Table 3.

Reviewer #1

This manuscript reports findings that nicely follow up the previous reports of this group on cellular metabolic pathways that play roles in cartilage homeostasis and/or dysfunction. Of interest is that ZFP36L1, among the RNA binding proteins of the ZFP36 family, appears to have a specific role in OA chondrocytes. The mechanism that the authors seem to be proposing is that upregulation of ZFP36L1 in OA cartilage could promote OA by upregulating apoptosis in cartilage through down-regulating HSPA1A, a chondroprotective intracellular protein. The authors also showed that ZFP36L1 promotes cartilage destruction possibly by indirect effects on IL-1 β -induced matrix-degrading proteins, but its ablation also attenuates osteophyte formation and subchondral bone plate thickness, which are not regulated by HSPA1A. Thus, these studies define a novel mechanism involving specific regulation by an RNA binding protein, but more work may be needed to determine how to precisely regulate this pathway for therapeutic intervention.

Comment-1. Introduction: It is unclear why the authors decided to specifically assess RNA binding proteins by microarray analysis in the context of OA pathogenesis. Is this because the class of genes appeared in their previous studies (e.g., in relation to ZIP8), or are there previously published studies indicating that RNA binding proteins may play a role in OA?

The expression of OA-related catabolic and anabolic factors, such as extracellular matrix molecules in cartilage and matrix-degrading enzymes in chondrocytes, can be regulated at multiple steps, including the post-transcriptional modulation of mRNA stability. This led us to examine whether OA pathogenesis is associated with post-transcriptional regulation of the stability of mRNAs encoding OA-related catabolic and anabolic factors. To this end, we performed microarray analysis of primary-culture mouse chondrocytes subjected to stimulation of OA-associated catabolism, such as by IL-1 β treatment or overexpression of ZIP8 or HIF-2 α . Specifically, we looked for upregulation of 114 genes whose encoded factors are involved in catabolism of mRNA. From among the examined genes (Table S1), we found that ZFP36L1, which is an AU-rich element (ARE)-binding protein, was specifically upregulated by all of the tested OA-associated catabolic regulators (Figure 1a). To address the reviewer's comment, we modified the Introduction section to more clearly indicate this aspect of our study (p. 3, 2nd paragraph) in our revised manuscript.

Comment-2. In the concluding two paragraphs of the Discussion, it is unclear whether both ZFP36L1 and HSPA1A should be considered the therapeutic targets. Since HSPA1A appears to play a protective role, it might be a therapeutic molecule itself, whereas ZFP36L1 could be a target because its deletion abrogates OA in mice. Several statements are confusing in that regard, for example, where it is stated that ZFP36L1 regulates mediators responsible for cartilage destruction, etc., and downstream targets of HSPA1A regulate mediators of cartilage destruction. Please consider rewriting these concluding paragraphs to make them less confusing.

We appreciate this comment and have revised the Discussion to clearly indicate that ZFP36L1 is a potential therapeutic target of OA and that HSPA1A appears to play a protective role in OA pathogenesis.

Comment-3. Methods, Page 15, lines 410/411: Subchondral bone sclerosis is a result of complex changes in tissue structure, organization, and mineralization. What is really being measured is subchondral bone plate thickness, which is correctly used in the labels on the graphs, so this terminology should be used in describing the results for Figure 2 (page 6, line 149), Figure 3 (page 7, lines 167 and 172), and Figure 5 (page 9, line 217) and in the Discussion (line 359, 361, etc.).

We agree with this comment. As indicated by the reviewer, subchondral bone sclerosis is a result of complex changes in tissue structure, organization, and mineralization. In this study, we measured the thickness of the subchondral bone plate (SBP) in an attempt to assess subchondral bone sclerosis, as previously reported by Das Neves Borges et al. (ref. 27). In the revised manuscript, we use the terminology “subchondral bone plate thickness” in the Methods, Results, Discussion, and Figure legends.

MINOR: Figure 6: correct “ADATS5” to “ADAMTS5”.

We modified revised Figure 6a accordingly.

Reviewer #2

An interesting study implicating the RNA-binding protein ZFP36L1 as a mediator of processes involved in the development of osteoarthritis. This is timely as there is increasing interest in the role of post-transcriptional and protein translational control as mediators of this disease. The study uses a number of in vivo and in vitro approaches to come to its conclusion. I do feel there are some issues that need to be addressed however.

Comment-1. (1) Primary chondrocytes de-differentiate quickly in culture and so detail needs to be provided regarding the human and murine primary chondrocyte work. Specifically, this should include the time in culture of the cells before they were used (e.g. passage number) and the quantitative expression of marker genes in these cells. (2) Building on this, in conditions used to model osteoarthritis responses (e.g. IL-1 β stimulation, or adenoviral overexpression of HIF-2 α and ZIP8) both human and murine systems need to be properly characterized by determining positive regulation of genes such as MMP3 at a quantitative level or by examining increased enzyme activity.

(1) As pointed out by the reviewer, primary chondrocytes rapidly de-differentiate in 2-dimensional culture. Indeed, we previously demonstrated that protein kinase C and extracellular signal-regulated protein kinase (ERK) regulate the de-differentiation of chondrocytes (Yoon et al., J. Biol. Chem. 277, 8412-20, 2002). We thus used passage 0 chondrocytes, as clearly indicated in the Results section (p. 5, 2nd paragraph) and Methods section (p. 18, 2nd paragraph) of the manuscript. To address the reviewer's comment, we additionally confirmed that passage 0 chondrocytes maintain their differentiated phenotypes, as evidenced by high expression levels of type II collagen and low expression levels of type I collagen. These additional data are presented in Fig. S1a and described in the Results section (p. 5, 2nd paragraph) of the revised manuscript.

(2) To address this issue, we detected and quantified the mRNA levels of MMP3 in chondrocytes treated with IL-1 β or infected with Ad-HIF-2 α or Ad-ZIP8. The additional data are presented in Figure S1b and S1c and described in the Results section (p. 5, 2nd paragraph) of the revised manuscript.

Finally, we used primary-culture mouse (not human) articular chondrocytes. We therefore characterized the mouse chondrocytes, as described above. However, we used human cartilage tissue (i.e., Figure 1d) to confirm that ZFP36L1 is upregulated in human OA chondrocytes.

Comment-2. (1) ZFP36L1 protein levels are shown to be higher in both human OA cartilage and in mouse OA model systems using immunohistochemistry. For the human tissue, shown in figure 1c, there is now description of whether the micrographs shown represent anatomically equivalent regions of the joint surface (e.g. location on surface and depth of tissue). This is important as different locations might see different levels of gene/protein expression. Full depth sections from equivalent joint regions should be presented with better histological characterization of proteoglycan loss and tissue fibrillation. In the same figure, the mouse OA model shows more clearly an increase in ZFP36L1 in the cartilage from DMM-treated mice or those injected with HIF-2 α or ZIP8 adenovirus.(2) In the controls in these cases there appears to be very little ZFP36L1. However, in the KO mouse analysis shown in supplementary figure 2c there is a lot of ZFP36L1 visible in the cartilage of control mice (I'm assuming they are controls although I think they are incorrectly labelled

‘+/-’ in supplementary figure 2c). On the surface there appears to be a mixed message given by these different figure with low expression being shown in control cartilage in figure 1 but high expression portrayed in supplementary figure 2C.

(1) As presented in Figure 1e of the revised manuscript, ZFP36L1 is markedly upregulated in chondrocytes of articular cartilage regions (above the tidemark) in all of the examined mouse OA models (i.e., those caused by DMM or overexpression of HIF-2 α or ZIP8), and lower protein levels of ZFP36L1 are detected in chondrocytes of calcified cartilage. To examine the expression levels of ZFP36L1 in human OA chondrocytes, we examined chondrocytes from the superficial zone to the deep zone, and found that ZFP36L1 proteins are detected in all zones. To show this expression pattern of ZFP36L1, we replaced the images shown in original Figure 1d with the new images presented in revised Figure 1d.

(2) To examine the expression levels of ZFP36L1 in cartilage sections of *Zfp3611*^{+/+} and *Zfp3611*^{+/-} mice (Figure S4c), we had to use a longer developing time during the immunohistochemical analysis, compared with that used to detect ZFP36L1 in cartilage sections of mice IA injected with Ad-C or Ad-ZFP36L1 (Figure 1e and Figure 2a). This might cause confusion regarding the basal levels of ZFP36L1 in normal mouse cartilage tissue. We thus replaced the images shown in original Figure S2c with the new and more representative images presented in revised Figure S4c.

Comment-3. (1) The use of the knockout mouse model does not appear to me to be totally conclusive. These mice are stated as having no developmental defects, but the evidence for this is limited to the skeletal staining in supplementary figure 2D. This analysis needs to be far more rigorous. Measurement of the length of skeletal elements during development and post-natally as well as histological confirmation that development remains the same is required. (2) It is odd that the authors have not decided to use a conditional knockout system to specifically determine the consequence of loss of ZFP36L1 in cartilage. This could complement the heterozygous null mouse and determine whether the role of the gene in OA is due to its function in cartilage or in other tissues.

(1) In addition to the skeletal staining of E18.5 embryos, we characterized *Zfp3611*^{+/-} mice more rigorously during our revision process. We found no difference in the lengths of the metatarsal bones of E18.5 *Zfp3611*^{+/-} mice and WT littermates (Figure S4e). We also performed Alcian blue/nuclear fast red staining of metatarsal bones from 2-week-old *Zfp3611*^{+/-} mice and WT littermates and found no significant difference in the lengths of the proliferative/resting and hypertrophic zones of the growth plates in the studied mouse groups (Figure S4f). These additional results are presented in Figure S4e and S4f and described in the corresponding portion of the Results section in the revised manuscript.

(2) We agree with the reviewer’s comment that a chondrocyte-specific conditional KO (cKO) system might provide valuable insights regarding the role of ZFP36L1 in OA pathogenesis. Although ES cells that may be used to generate *Zfp3611* cKO mice are now available from the European Mouse Mutant Cell Repository (these cells were not available when we began this study), it is likely to take an additional couple of years to generate cKO mice. Moreover, it is technically impossible to generate mice with specific cKO in all cell types of joint tissues. Thus, rather than generating cKO mice, we used whole-body deletion (*Zfp3611*^{+/-}) and whole-joint knockdown of ZFP36L1 via IA injection of

Ad-shZFP36L1. Importantly, OA is a whole-joint disease that involves multiple pathological changes in all joint tissues. Therefore, we believe that *Zfp361l*^{+/-} mice and whole-joint knockdown of ZFP36L1 are more appropriate models for elucidating OA as a whole-joint disease. We now discuss these points in the revised manuscript (p. 13, 2nd paragraph).

Comment-4. The use of adenoviruses to overexpress the genes of interest (ZFP36L1 and HSPA1A) or in some cases to express shRNAs targeting these genes is used a great deal in this study and appears to lead to strong infection and expression in the articular chondrocytes. There is very little focus however on the degree to which other joint tissues are being infected and thus expressing these factors. Evidence for modulation of ZFP36L1 and HSPA1A should be measured in the synovium, cruciate ligament and meniscus as well.

We previously reported that ~70% of chondrocytes in articular cartilage are infected by IA injection of Ad-GFP (ref. 26). To address the reviewer's point, we performed immunohistochemistry on sections of whole-joint tissues (cartilage, meniscus, synovium, and ligament) in mice IA injected with Ad-ZFP36L1 or Ad-HSPA1A. Similar to our previous findings with respect to HIF-2 α (Nat Med. 16, 687-93. 2010) and ZIP8 (Cell. 156, 730-43. 2014), the target molecules were effectively overexpressed not only in cartilage tissues but also in meniscus, synovium, and ligament. These additional results are presented in Figure S2a (Ad-ZFP36L1) and Figure S7a (Ad-HSPA1A) and described in the corresponding portion of the Results section in the revised manuscript.

Comment-5. Throughout the manuscript there is inconsistency in the gene expression analysis of the in vitro experiments regarding the use of end point or quantitative real time PCR. The latter should be used in all cases. Conclusions such as those for figure 2C where it is stated that “ZFP36L1 overexpression in primary-culture chondrocytes did not affect the expression levels of MMP3, MMP13, and ADAMTS5 which play crucial roles in OA cartilage destruction, nor did it affect cartilage ECM molecules, such as type II collagen and aggrecan, or the regulatory transcription factor, SOX9” cannot be justified by the use of end point PCR alone.

We addressed this issue by subjecting the RT-PCR results presented in original Figures 1b and 2c to qRT-PCR analysis, the results of which are now presented in revised Figure 1b and revised Figure S1c. Similarly, our qRT-PCR analysis of the RT-PCR results presented in original Figure 2c are now presented in revised Figure S2b. As a positive control for our qRT-PCR analysis, we exposed chondrocytes to IL-1 β and quantified the expression levels of MMP3, MMP13, ADAMTS5, SOX9, Coll-II, and aggrecan (Figure S2b).

Comment-6. Is there a fundamental reason why HIF-2 α and ZIP8 adenoviruses need to be respectively called Ad-*Epas1* and Ad-*Slc39a8*. Surely names like “Ad-Hif2 α ” and “Ad-Zip8” would result in far less confusion for the reader?

We used Ad-*Epas1* and Ad-*Slc39a8* to indicate the formal gene names (*Epas1* for HIF-2 α and *Slc39a8* for ZIP8). However, we agreed with the reviewer that this could cause confusion for the readers. Accordingly, we have changed Ad-*Epas1* to Ad-HIF-2 α and Ad-*Slc39a8* to Ad-ZIP8 in the revised manuscript.

Comment-7. The authors focused on the expression of HSPA1A as there were only 6 nucleotides differences between it and the HSPA1B gene. Is this within the whole mRNA (including 3-UTR) or just the coding region? If the former, then wouldn't study of the other gene also be useful as the two may exhibit distinct regulatory differences through different 3-UTR elements.

We appreciate the reviewer's valuable comment. The coding regions of murine HSPA1A and HSPA1B differ by only 10 nucleotides, and the encoded proteins are almost identical, as they differ by only one amino acid. We found considerable differences in the 3'-UTR sequences of HSPA1A and HSPA1B (52.6% homology), suggesting that they may undergo differential regulations. However, our microarray analysis indicated that HSPA1A and HSPA1B undergo similar regulatory changes in our system (revised Figure 4a). During the revision process, we additionally demonstrated that the mRNA levels of HSPA1A and HSPA1B are similarly upregulated by knockdown of ZFP36L1 via Ad-shZFP26L1 infection (revised Figure 4b, c). We also found that, similar to HSPA1A, HSPA1B contains several AREs in its 3'-UTR (Figure S6c). This supports the notion that both HSPA1A and HSPA1B are regulated by ZFP36L1. Our additional results are presented in Figures 4b, 4c, S6a, and S6c, described in the corresponding portion of the Results section (p. 8, 2nd paragraph), and addressed in the Discussion section (p.14, 1st paragraph and p. 15, 1st paragraph) of the revised manuscript.

Comment-8. Figure 5C HSPA1A adenovirus in DMM model. The image for the DMM + HSPA1A-Ad does not illustrate the data very well as there appears to be a significant reduction in subchondral bone sclerosis compared to the DMM Ad-C treatment, even though this parameter is not found to be significant in the scoring.

We addressed this issue by replacing the original images with more representative Safranin-O staining images in revised Figure 5c.

Reviewer #3

The manuscript finds a role for the RNA binding protein ZFP36L1 and the chaperone HSP70 (HSPA1A) in osteoarthritis. The authors use primary mouse chondrocytes, and an in vivo mouse model of surgical disruption of the knee joint (DMM). Treatments (eg. IL-1 β) that promote breakdown of cartilage upregulated ZFP36L1 in chondrocytes, human patient samples and the mouse model. While ZFP36L1 overexpression did not cause any pathology, depletion in the ZFP36L1^{+/-} mouse or by knockdown, partially relieved the disease phenotype. In chondrocytes, HSPA1A and HSPA1B transcripts were the most highly upregulated upon ZFP36L1 knockdown. ZFP36L1 associated with one region of ARE's in the HSPA1A 3'-UTR, and overexpression of ZFP36L1 increased the turnover of HSPA1A transcript. In the DMM model, overexpression of HSPA1A in the joint partially relieved the loss of cartilage, although other phenotypes were not affected. HSPA1A also prevented apoptosis of chondrocytes in articular cartilage after DMM surgery, and in cultures of chondrocytes. It is proposed that ZFP36L1 regulation of HSPA1A may be a therapeutic target.

The findings are interesting, and reveal an unexpected mode of HSPA1A regulation. A role for ZFP36L1 has not previously been known in chondrocytes and osteoarthritis, and the results in the heterozygous mouse show its importance for the disease. HSPA1A is well established to protect against apoptosis under various conditions. Chondrocyte apoptosis is part of the osteoarthritis process and the manuscript now suggests a mechanism that regulates it.

However, the authors do not take the last step and present a full disease mechanism, that high ZFP36L1 expression in osteoarthritis causes suppression of HSPA1A, allowing apoptosis to proceed. Overexpression of ZFP36L1 in healthy mice did not have much effect, the explanation put forward is that HSPA1A levels are low in the non-stressed condition.

Comment-1. Show the levels of HSPA1A protein expression in the DMM tissue, for the wild-type and ZFP36L1^{+/-} mouse, or with its knockdown. The data showing ZFP36L1 regulation of HSPA1A are only in chondrocytes in Fig. 4. Supplemental Fig. 4 shows some expression of HSPA1A in response to HIF-2 α and ZIP8, suggesting there is some expression in the disease state with wild-type ZFP36L1 levels.

We appreciate the reviewer's valuable comment, and addressed this issue by performing additional immunohistochemical staining of HSP70. We found that the basal protein levels of HSP70 in the cartilage of sham-operated mice were decreased upon DMM surgery. Knockdown of ZFP36L1 (via IA injection of Ad-shZFP36L1 or in *Zfp36l1*^{+/-} mice) or IA injection of Ad-HSPA1A in cartilage of sham-operated mice caused marked upregulation of HSP70 protein levels, and these levels remained elevated upon DMM surgery. These results collectively support our proposal that elevation of HSP70 proteins in cartilage is associated with the inhibition of DMM-induced OA pathogenesis. These additional results are presented in Figures S7b (*Zfp36l1*^{+/-}), S7c (Ad-shZFP36L1), and S7d (Ad-HSPA1A), and described in the corresponding portion of the Results section in the revised manuscript.

Comment-2. In Fig. 4b/4c in chondrocytes, show the protein levels of HSPA1A by Western blot. A 10-fold change in mRNA does not always mean that much difference in protein expression. It would

be preferable to show ZFP36L1 levels by Western blot too. The effect on HSPA1A protein should also be addressed in a ZFP36L1 co-expression experiment like in Figure 4e.

We addressed this issue by performing Western blot analysis, the results of which are presented in Figures 4c and 4e of the revised manuscript. We found that knockdown of ZFP36L1 via infection of Ad-shZFP36L1 increased HSP70 at the mRNA and protein levels, while overexpression of ZFP36L1 via infection of Ad-ZFP36L1 decreased HSP70 at the mRNA and protein levels (Figure 4c). Additionally, overexpression of ZFP36L1 in HSP70-overexpressing chondrocytes also decreased HSP70 expression in primary-culture chondrocytes.

Comment-3. Show articular chondrocyte apoptosis is decreased in the *Zfp361l*^{+/-} mouse and with its knockdown, comparable to Fig. 6d.

As suggested by the reviewer, we identified and quantified apoptotic chondrocytes in cartilage tissues of *Zfp361l*^{+/-} mice or wild-type mice IA injected with Ad-shZFP36L1 upon sham operation or DMM surgery. We report here that, similar to the overexpression of HSPA1A, knockdown or knockout of ZFP36L1 significantly inhibited the apoptosis of chondrocytes in DMM-operated articular cartilage. These additional results are presented in Figures 6e and 6f and described in the Results section (p. 11, 1st paragraph) of the revised manuscript.

Comment-4. Knockdown HSPA1A in the *Zfp361l*^{+/-} mouse to show that it restores the disease phenotype after DMM surgery.

We appreciate the reviewer's valuable suggestion, which we addressed by performing additional experiments. We knocked down HSPA1A in WT and *Zfp361l*^{+/-} mice via IA injection of Ad-shHSPA1A in sham- or DMM-operated mice. Our results revealed that IA injection of Ad-shHSPA1A effectively inhibited the increase of HSP70 protein in the cartilage of DMM-operated *Zfp361l*^{+/-} mice. Consistent with this, knockdown of HSPA1A in *Zfp361l*^{+/-} mice restored OA cartilage destruction. Our results clearly support our contention that upregulation of HSP70 is responsible for the inhibitory effects of ZFP36L1 knockdown on cartilage destruction. In contrast, ZFP36L1 knockdown had no significant effect on DMM-induced osteophyte formation or subchondral bone plate thickness. These results are essentially the same as those observed following HSPA1A overexpression. Our additional results are included in Figure 7, described in the Results section (p. 11, 2nd paragraph~p. 12, 1st paragraph), and addressed in the Discussion section (p. 15, 2nd paragraph and p. 16, 1st paragraph) of the revised manuscript.

Comment-5. A catalytically inactive mutant of HSPA1A (e.g. K71E) is a better control for the overexpression experiments in Fig. 6.

As suggested by the reviewer, we overexpressed K71E HSPA1A, which is a mutant that lacks ATPase activity and chaperone functions, and examined SNP-induced chondrocyte apoptosis. We found that whereas wild type (WT)-HSPA1A protects chondrocytes from SNP-induced apoptosis, K71E-HSPA1A did not modulate this process. These findings indicate that the chaperone function of HSPA1A is necessary for its inhibitory effects on chondrocyte apoptosis. The additional results are

presented in Figure 6c and described in the Results section (p. 10, 2nd paragraph) of the revised manuscript.

Reviewers' Comments:

Reviewer #1:

Remarks to the Author:

I do not have any additional comments.

Reviewer #2:

Remarks to the Author:

The authors have made a considerable number of changes to the manuscript and, on the whole, have addressed the comments that I have made.

The lack of a conditional cartilage knockout (or knockdown) of ZFP36L1 continues to be a weakness. The authors are correct that multiple tissues within the joint region contribute to osteoarthritis, but this paper focuses mainly on a role for ZFP36L1 in chondrocytes. Whilst it seems likely that the effects that they see are mediated through chondrocyte centred mechanisms, without evidence from animals with cartilage-specific conditional disruption of the gene there is still the possibility that ZFP36L1 modulates osteoarthritis through its action in other tissues (e.g synovium, bone). It is a shame that given the extensive amount of work presented in the paper, this key question remains unresolved at present.

Reviewer #3:

Remarks to the Author:

The revisions have addressed my concerns. The pathway is now shown in cells and in the disease model in mice. These are interesting findings which are experimentally supported.

Point-to-point response to reviewers' comments

Reviewer #1 (Remarks to the Author):

I do not have any additional comments.

[Response] We thank the reviewer for these kind comments.

Reviewer #2 (Remarks to the Author):

The authors have made a considerable number of changes to the manuscript and, on the whole, have addressed the comments that I have made. The lack of a conditional cartilage knockout (or knockdown) of ZFP36L1 continues to be a weakness. The authors are correct that multiple tissues within the joint region contribute to osteoarthritis, but this paper focuses mainly on a role for ZFP36L1 in chondrocytes. Whilst it seems likely that the effects that they see are mediated through chondrocyte centered mechanisms, without evidence from animals with cartilage-specific conditional disruption of the gene there is still the possibility that ZFP36L1 modulates osteoarthritis through its action in other tissues (e.g., synovium, bone). It is a shame that given the extensive amount of work presented in the paper, this key question remains unresolved at present.

[Response] We addressed this issue by modifying our Discussion to (p. 13): “The use of a chondrocyte-specific conditional KO (cKO) system, instead of whole-body heterozygous KO, might provide valuable insights into the role of ZFP36L1 in chondrocytes during OA cartilage destruction. Because we focused mainly on a role for ZFP36L1 in chondrocytes, the limitation of current studies is the lack of chondrocyte-specific cKO system. However, OA is a whole-joint disease that involves multiple cell types and multiple pathological changes in all joint tissues^{2,3}. Therefore, whole-body deletion and whole-joint knockdown of a target gene may be alternative strategies than its cell-type-specific alteration when seeking to examine OA as a whole-joint disease.”

Reviewer #3 (Remarks to the Author):

The revisions have addressed my concerns. The pathway is now shown in cells and in the disease model in mice. These are interesting findings which are experimentally supported.

[Response] We thank the reviewer for these kind comments.